



# A 1500-year multiproxy record of coastal hypoxia from the northern Baltic Sea indicates unprecedented deoxygenation over the 20th century

Sami A. Jokinen[1], Joonas J. Virtasalo[2], Tom Jilbert[3], Jérôme Kaiser[4], Olaf Dellwig[4], Helge W. Arz[4], Jari Hänninen[5], Laura Arppe[6], Miia Collander[7], Timo Saarinen[1]

[1]Department of Geography and Geology, University of Turku, 20014 Turku, Finland
[2]Marine Geology, Geological Survey of Finland (GTK), P.O. Box 96, 02151 Espoo, Finland
[3]Department of Environmental Sciences, University of Helsinki, P.O. Box 65, 00014 Helsinki, Finland
[4]Leibniz Institute for Baltic Sea Research Warnemünde (IOW), Seestrasse 15, 18119 Rostock, Germany
[5]Archipelago Research Institute, University of Turku, 20014 Turku, Finland
[6]Finnish Museum of Natural History, University of Helsinki, P.O. Box 64, 00014 Helsinki, Finland
[7]Department of Food and Environmental Sciences, University of Helsinki, P.O. Box 66, 00014 Helsinki, Finland

*Correspondence to*: Sami A. Jokinen (sami.jokinen@utu.fi)



**Abstract.** The anthropogenically forced expansion of coastal hypoxia is a major environmental problem affecting coastal ecosystems and biogeochemical cycles throughout the world. The Baltic Sea is a semi-enclosed shelf sea whose central deep basins have been highly prone to deoxygenation during its Holocene history, as shown previously by numerous paleoenvironmental studies. However, long-term data on past fluctuations in the intensity of hypoxia in the coastal zone of the Baltic Sea are largely lacking, despite the significant role of these areas in retaining nutrients derived from the catchment. Here we present a 1500-year multiproxy record of near-bottom water redox changes from the coastal zone of the northern Baltic Sea, encompassing the climatic phases of the Medieval Climate Anomaly (MCA), the Little Ice Age (LIA), and the Modern Warm Period (MoWP). Our reconstruction shows that although multicentennial climate variability has modulated depositional conditions and delivery of organic matter (OM) to the basin the modern aggravation of coastal hypoxia is unprecedented, and besides gradual changes in the basin configuration, it must have been forced by excess human-induced nutrient loading. The progressive deoxygenation since the beginning of 1900s was originally triggered by the combined effects of gradual shoaling of the basin and warming climate, which amplified sediment focusing and increased the vulnerability to hypoxia. Importantly, the anthropogenic eutrophication of coastal waters in our study area began decades earlier than previously thought, leading to a marked aggravation of hypoxia in the 1950s through fueling primary productivity, while we find no evidence of anthropogenic forcing during the MCA. These results have implications for the assessment of reference conditions for coastal water quality. Furthermore, this study highlights the need for combined use of sedimentological, ichnological, and geochemical proxies in order to robustly reconstruct subtle redox shifts especially in dynamic, non-euxinic coastal settings with strong seasonal contrasts in the bottom water quality.





# 1 Introduction

The expansion of hypoxic dead zones is an ongoing global problem both in the marine realm (Diaz and Rosenberg, 2008; Vaquer-Sunyer and Duarte, 2008; Gooday et al., 2009; Rabalais et al., 2010; Rabalais et al., 2014) as well as in lacustrine settings (Jenny et al., 2016a, b). Bottom water oxygen depletion (< 2 mg L$^{-1}$ dissolved oxygen), caused by the combined effects

of water column stratification and excess delivery of organic matter (OM) to the seafloor, deteriorates benthic ecosystems (Levin et al., 2009) and often triggers harmful algal blooms (Zhang et al., 2010) due to the impact of hypoxia on biogeochemical cycles at the sediment–water interface (Middelburg and Levin, 2009). Upon bottom water deoxygenation, phosphorus (P) is released efficiently to the water column from surface sediments, fueling further primary productivity and dinitrogen (N$_2$) fixation by diazotrophic cyanobacteria, thus triggering a self-sustaining positive feedback mechanism

commonly associated with eutrophication (Vahtera et al., 2007). In addition, the ability of benthic ecosystems to remove nitrogen via denitrification and anaerobic ammonium oxidation may be reduced upon repeated or prolonged exposure to bottom water hypoxia (Conley et al., 2009a; Middelburg and Levin, 2009; Carstensen et al., 2014b). Due to these internal feedback mechanisms, recovery from hypoxia is often slow, hampering management of the problem through reductions in external nutrient loading (Vahtera et al., 2007). Furthermore, global warming is likely to exacerbate the spreading of hypoxia

in many regions through enhanced nutrient inputs (linked to increased precipitation and discharge), decreased solubility of oxygen due to increased temperature, and acceleration of internal nutrient cycling (Meier et al., 2011; Meire et al., 2013).

Over the past century, the Baltic Sea has seen a marked expansion of benthic hypoxia (Jonsson et al., 1990; Conley et al., 2011: Carstensen et al., 2014a), and the Baltic Sea dead zone is often referred to as the largest anthropogenically induced hypoxic marine area in the world (Diaz and Rosenberg, 2008). Yet, although long-term trends in the expansion of hypoxia in

offshore areas of the Baltic Sea have been widely studied, little is known about the past evolution of hypoxia in the shallow coastal areas, where episodic or seasonal oxygen deficiency is forced by thermal rather than salinity stratification (Virtasalo et al., 2005; Conley et al., 2011). Importantly, these coastal areas act as a filter for nutrients received from the catchment (Asmala et al., 2017). Thus, changes in biogeochemical cycles in coastal sediments may impact on nutrient transport to offshore areas of the Baltic Sea (Almroth-Rosell et al., 2016). Elucidating fluctuations in coastal hypoxia will aid understanding of the

efficiency of the coastal filter, and is therefore vital for understanding the expansion of hypoxia in the entire Baltic Sea. Indeed, it is still debated whether the decisive factor triggering the hypoxic event during the Medieval Climate Anomaly (MCA, 900–1350 AD) in the Baltic Proper was intensified land use in the catchment (Zillén and Conley, 2010) or anomalously warm climate (Kabel et al., 2012; Papadomanolaki et al., 2018).

In this study, we present a multi-proxy reconstruction of the development of hypoxia in an enclosed coastal setting in the

Finnish Archipelago Sea (northern Baltic Sea) over the past 1500 years, covering the known climatic oscillations of the MCA, the Little Ice Age (LIA, 1350–1850 AD) and the Modern Warm Period (MoWP, after 1850 AD) in order to assess how the



coastal zone responds to centennial–millennial climate variability and potential past inputs of nutrients from the catchment areas. We use diverse bulk sediment geochemical proxies in combination with integrated sedimentological and ichnological analyses to elucidate temporal changes in the intensity of near-bottom water oxygen deficiency. In order to constrain the drivers behind the observed oxygenation changes, we assess past fluctuations in hydrodynamic conditions at the study site, and in the

delivery of OM, and compare these with the past climate variability and changes in the anthropogenic nutrient loading from the catchment.

## 2 Study location

The Baltic Sea is a shallow (mean depth 54 m) semi-enclosed basin located on a continental shelf (Fig. 1a) between maritime temperate and continental sub-Arctic climate zones. Climatic conditions in the area are largely modulated by the North Atlantic

Oscillation (NAO) as well as the summer low and winter high over Eurasia (e.g. Rutgersson et al., 2014). Winter mean air temperature ranges from -12 ℃ in the north to 0 ℃ in the south, whereas summer mean temperature has a narrower range of 14–17 ℃. The sea is essentially non-tidal, but irregular variations in wind and atmospheric pressure modulate the water level with maximum amplitude of 2 m.

Surface salinity exhibits an increasing trend from north to south, from 3–5 in the Gulf of Finland and Gulf of Bothnia to 8–10

in the southern Baltic (Leppäranta and Myrberg, 2009). This horizontal salinity gradient results from combined effects of high riverine freshwater input in the north and occasional inflows of saline water from the North Sea through the Danish Straits in the south. The contrasting density of these two water masses leads to formation of a strong 10–20 m thick pycnocline, which lies at a depth of 40–80 m depending on the sub-basin (Leppäranta and Myrberg, 2009). Irregular saline inflow events from the North Sea occasionally ventilate the deep stagnant bottom waters of the Baltic Proper, but this oxygen is readily exhausted

with a net effect of stronger stratification and possibly even more severe oxygen depletion (Conley et al., 2002; Carstensen et al., 2014a).

The Archipelago Sea, located in the south-western coastal area of Finland in the northern Baltic Sea (Fig. 1a), is a mosaic of thousands of islands and small bays within an area of ~ 8000 km$^2$. Salinity in the area ranges from 5 to 7, increasing towards the open sea. Mean water depth is only 23 m, although some deeps reach over 100 m. The length of the ice season is 3–4.5

months (Seinä, 1994), but a current decreasing trend of 46 days per century has been reported (Ronkainen, 2013). The rate of the present glacio-isostatic uplift is 3–4 mm per year (Mäkinen and Saaranen, 1998), which exposes previously deposited sediments to wave erosion and modulates hydrographic conditions in the area.

The complex topography of the Archipelago Sea results in restricted water exchange between the inner archipelago and the open-sea areas (Mälkki et al., 1979), and numerous small basins with contrasting bottom water conditions exist in close





proximity (Virtasalo et al., 2005). Water exchange in the area mostly occurs through deep straits following the fault-lines of the crystalline bedrock, which are mostly aligned in a north-south direction. In enclosed basins, a strong thermocline impedes mixing of dissolved oxygen to the bottom waters during summer, which together with high delivery of reactive OM to the seafloor commonly results in seasonal hypoxia (Virtasalo et al., 2005; Jokinen et al., 2015). Mixing of the water column

through thermal convection takes place in spring and autumn due to the lack of a permanent halocline (Leppäranta and Myrberg, 2009).

The sediment fill of the Archipelago Sea since the deglacial to present comprises a succession of ice-proximal tills and outwash, glaciolacustrine rhythmites, patchily-distributed debrites, postglacial lacustrine clays, and brackish-water mud drifts (Virtasalo et al., 2007; 2014). The study area was deglaciated at ~ 11400 cal. BP (Stroeven et al., 2016), which led to freshwater conditions

in the area (Tuovinen et al., 2008). By ~ 7600 cal. BP, the eustatic ocean-level rise surpassed the glacio-isostatic rebound rate and the Danish Straits became inundated, resulting in marine flooding of the Baltic Sea basin and consequent salinization of the Archipelago Sea (Tuovinen et al., 2008). Since then, sedimentation in the Archipelago Sea has been characterized by wave and current modulated deposition of brackish-water mud drifts with laminated intervals and local unconformities (Virtasalo et al., 2007).

Haverö is a small, extremely enclosed basin in the middle of the Archipelago Sea (Fig. 1). The Proterozoic bedrock surface in the area is dominated by microcline granites, with small patches of gneisses, amfibolites and granodiorite (Bedrock of Finland, DigiKP). Due to the relatively high elevation of the surrounding islands, Quaternary deposits have been largely removed by erosion after the islands were uplifted above the wave base (Maankamara, DigiKP). The distance to the mouths of the largest rivers in the area, Aurajoki River and Paimionjoki River, is 25 and 38 km, respectively. A small brook drains into the basin

from a lake located on the Haverö Island. Sedimentation in the Haverö basin is dominated by reworking of previously deposited late- and postglacial clays and organic-rich brackish-water muds during autumn and winter, and by rapid settling of organic-rich aggregates during spring and summer (Jokinen et al., 2015). This seasonal contrast in the sedimentation, accompanied by severe seasonal hypoxia and consequent deterioration of macrobenthic fauna, has enabled the formation and preservation of annual laminations (varves) over the past decades (Jokinen et al., 2015). Population around Haverö is sparse, and the dominant

direct anthropogenic nutrient loading is sourced from two local fish farming cages that were operational from 1987 to 2008 AD (Fig. 1b).



## 3 Description of the proxies

### 3.1 Proxies for hydrodynamic conditions

#### 3.1.1 Sediment grain size

Sediment grain size depends on sediment inputs and hydrodynamic conditions. Excluding river mouths, grain size distribution
in the coastal zone of the Baltic Sea is mainly governed by wind stress and basin morphometry that modulate the bottom water
energy flux (Lehmann et al., 2002b; Jönsson et al., 2005a; Ning et al., 2016). In general, periods with enhanced near-bottom
currents become recorded in sediments through increased proportion of coarse grains (e.g. Jönsson et al., 2005a).

#### 3.1.2 Titanium to potassium ratio

In fine-grained sediments, potassium (K) is mainly associated with illite $((K,H_3O)(Al,Mg,Fe)_2(Si,Al)_4O_{10}[(OH)_2(H_2O)])$,
whereas titanium (Ti) is mainly present in heavy minerals such as rutile $(TiO_2)$ or ilmenite $((Fe, Mg, Mn, Ti)O_3)$ and thus it
concentrates in the coarse fraction of sediments due to sorting effects (Dellwig et al., 2000). Therefore, variations in Ti/K in
locations with negligible changes in sediment provenance over time can be ascribed to changes in depositional energy and
sediment transport processes (Piva et al., 2008; Spofforth et al., 2008), whereby an increase in the ratio denotes amplified
bottom water currents.

### 3.2 Proxies for the source of organic matter

#### 3.2.1 Carbon to nitrogen ratio

Due to the contrasting composition of vascular plants in comparison to phytoplankton, carbon (C) to nitrogen (N) ratios of
sediment OM can be used to estimate relative contributions of OM originating from terrestrial and marine compartments
(Meyers, 1994; 1997 and references therein). This difference arises from the high abundance of proteins in algae, while
vascular plants are rich in cellulose instead. Accordingly, marine and terrestrial OM is characterized by molar C/N ratios of
4−10 and > 20, respectively. However, C/N of OM produced in the euphotic zone is potentially elevated during sinking (Müller,
1977; Meyers, 2003) and diagenesis (Gälman et al., 2008) due to preferential degradation of N over C. Conversely, adsorption
of ammonia (produced upon OM decomposition) on to clay mineral surfaces has the potential to decrease sediment C/N during
diagenesis especially in sediments with less than 0.3 % of $C_{org}$ (Müller, 1977). Despite these potential constraints, C/N ratio
often accurately records past variations in the source of OM to the seafloor (Meyers, 1994; 1997), as suggested for the coastal
Baltic Sea (Müller and Mathesius, 1999).





### 3.2.2 Stable isotope composition of carbon

Stable isotope composition of sediment organic carbon ($\delta^{13}C_{org}$) depends on the isotopic ratio of the C source as well as on the fractionation between $^{12}C$ and $^{13}C$ during photosynthesis (e.g. Hayes 1993). A vast majority of plants present in the study area fix C via the Calvin (C$_3$) pathway, whereby $^{12}C$ is preferentially incorporated, producing C$_{org}$ with 20 ‰ lighter isotope

composition than the inorganic C (C$_{inorg}$) source. Land plants use atmospheric carbon dioxide (CO$_2$, $\delta^{13}C$ ~ -7 ‰) as their source of C$_{inorg}$, whereas marine algae utilize dissolved bicarbonate (HCO$_3^-$, $\delta^{13}C$ ~ 0 ‰) in addition to CO$_2$, resulting in higher $\delta^{13}C$ values in marine phytoplankton. Although the preferential source of C for marine algae is dissolved CO$_2$, they incorporate a progressively higher proportion of HCO$_3^-$ with respect to CO$_2$ under decreasing pCO$_2$ (Fogel et al., 1992; Rost et al., 2003), leading to increasing $\delta^{13}C$ values in their cells. Accordingly, organic matter produced by land plants has an average $\delta^{13}C$ value

of ~ -27 ‰, whereas phytoplankton-derived marine organic matter is characterized by a $\delta^{13}C$ signature from -20 to -22 ‰ (Meyers, 1994). Although early diagenetic processes potentially alter the original stable isotope composition of OM (Lehmann et al., 2002a), a number of studies indicate that the $\delta^{13}C$ signature of sediment OM often robustly records past environmental changes in the water column (Meyers, 1994, 1997; Freudenthal et al., 2001; Kohzu et al., 2011).

### 3.2.3 Stable isotope composition of nitrogen

The stable isotope composition of nitrogen ($\delta^{15}N$) in marine OM produced in the euphotic zone is governed by the isotopic ratio of nitrate (NO$_3^-$) and the extent to which this nutrient reservoir is consumed by phytoplankton (Altabet and Francois, 1994; Voss et al., 1996). Due to the preferential assimilation of $^{14}NO_3^-$ during photosynthesis, the produced OM is depleted in $^{15}N$ relative to the inorganic N source. If the available nutrient pool is not continuously replenished (e.g. due to thermal stratification), the $\delta^{15}N$ signature of the remaining NO$_3^-$ pool becomes progressively enriched following Rayleigh fractionation

kinetics (Altabet and Francois, 1994). Consequently, upon complete exhaustion of the NO$_3^-$ pool due to a period of intensive primary productivity, such as a spring bloom, no net isotopic fractionation occurs and the "original" $\delta^{15}N$ signature of NO$_3^-$ is communicated to the accumulated OM. On continental margin settings characterized by high sedimentation rates and intensive phytoplankton blooms that contribute the majority of OM delivery to the seafloor, as in the Baltic Sea, alteration of the primary $\delta^{15}N$ signal attained by phytoplankton within the euphotic zone is minute both during sinking through the water column

(Altabet et al., 1991; Altabet and Francois, 1994) and during early diagenesis (Kienast et al., 2002; Thunell et al., 2004). Hence, high $\delta^{15}N$ values in coastal sediments often record eutrophication through agricultural and urban contribution to riverine N loading (McClelland and Valiela, 1998, Voss et al., 2000; Struck et al., 2000; Cole et al., 2004), owing to the enrichment of $^{15}N$ in fertilizer and manure due to N transformations such as denitrification and ammonia volatilization in catchment soils (Heaton, 1986; Aravena et al., 1993).



### 3.2.4 Branched isoprenoid tetraether index

Founded on the observation that branched glycerol dialkyl glycerol tetraethers (GDGTs I-III) are mainly sourced from terrestrial environment (soil OM), whereas crenarchaeol originates predominantly from marine environment (a characteristic lipid for aquatic *Thaumarchaeota*), the branched isoprenoid tetraether (BIT) index was defined by Hopmans et al. (2004) as a proxy for the relative abundance of terrestrial OM:

$$BIT\ index = \frac{[GDGT\text{-}I] + [GDGT\text{-}II] + [GDGT\text{-}III]}{[Crenarchaeol] + [GDGT\text{-}I] + [GDGT\text{-}II] + [GDGT\text{-}III]}, \tag{1}$$

where end-member values of 0 and 1 denote open marine and coastal environment, respectively. Diagenetic effects on BIT index are regarded minute due to the structural similarity of the compounds involved (Schouten et al., 2013). Possible constraints on the application of the BIT index include potential in situ production of branched GDGTs in the water column (Sinninghe Damsté et al., 2009) and marine sediments (Peterse et al., 2009) as well as the potential production of crenarchaeol in soils (Weijers et al., 2006).

### 3.3 Redox proxies

### 3.3.1 Sedimentary-fabric

Sedimentary-fabric analysis is focused on the preservation of primary sedimentary structure, its mixing by macrofaunal bioturbation, and the characteristics of identifiable bioturbation structures (trace fossils). Trace fossil assemblages of the organic-rich brackish-water muds of the Baltic Sea are well applicable for reconstructing past bottom water redox shifts (Virtasalo et al., 2011a, b). Benthic faunal responses to bottom water hypoxia include avoidance or even mortality of large species, loss of diversity, and shoaling of penetration depth or emergence from sediment (Levin et al., 2009). Consequently, the vertical extent, diameter, and diversity of burrows constructed by macrobenthic fauna decrease with declining bottom water oxygenation, which is thought to be the decisive factor shaping biogenic sedimentary-fabrics in the Baltic Sea (Savrda and Bottjer, 1986, 1991; Virtasalo et al., 2011a, b), although other factors such as salinity, substrate consistency, and food supply also affect trace fossil assemblages in the area (Virtasalo et al., 2006, 2011a). Importantly, the behavior of macrobenthic fauna responds rapidly to changes in the bottom water environmental conditions, and these responses can be readily recorded in the trace fossil assemblages (Savrda and Bottjer, 1986; Wetzel, 1991). The magnitude of this response is governed by the intensity and duration the deoxygenation as well as by the recovery time between consecutive hypoxic events (Levin et al., 2009).

### 3.3.2 Molybdenum content

Sedimentary molybdenum (Mo) content is a well-established proxy for past redox fluctuations in bottom waters overlying marine sediments (e.g. Algeo and Lyons, 2006; Scott and Lyons, 2012; Helz and Adelson, 2013). It has been successfully




applied to Baltic Sea sediments for bottom water redox reconstructions, especially in deep areas (Mort et al., 2010; Jilbert and Slomp, 2013; Jilbert et al., 2015; Dijkstra et al., 2016; Hardisty et al., 2016; van Helmond et al., 2017). The sensitivity of sedimentary Mo content to redox fluctuations is due to the conversion of the relatively inert molybdate ion ($MoO_4^{2-}$) in seawater to a series of particle-reactive thiomolybdates ($MoO_xS_{4-x}$) under exposure to hydrogen sulfide ($H_2S$). Where the concentration

of $H_2S_{aq}$ exceeds ~ 11 µM, the so-called sulfide-switch is activated and a quantitative conversion to tetrathiomolybdate $MoS_4^{2-}$ may occur (Helz et al., 1996; Erickson and Helz, 2000), triggering effective fixation of Mo in association with Fe-S phases (Helz, et al., 1996, 2011; O´Connor et al., 2015) and organic matter (Helz et al., 1996; Algeo and Lyons 2006; Dahl et al., 2017). In addition, Mo is reduced from oxidation state (VI) to (IV) during burial in sediments (Dahl et al., 2013). Under non-euxinic conditions, where $H_2S$ is only found in pore waters but not in bottom waters, the initial sedimentation of water column

Mo often occurs through adsorption to solid-phase manganese (Mn) oxides at the sediment–water interface (Scott and Lyons, 2012; Scholz et al., 2013; Noordmann et al., 2015). Upon reduction of Mn oxides, Mo is released into the pore waters, from where it may efflux back to the water column or become sequestered into the sediments in the presence of sufficiently high sulfide levels. In such settings, the depth and intensity of the pore water $H_2S$ maximum (in Baltic Sea sediments typically associated with the sulfate–methane transition zone SMTZ, Egger et al., 2015; Jilbert et al., 2017) is expected to regulate the

amount of Mo sequestered in the sediment (Adelson et al., 2001; Scott and Lyons, 2012; Helz and Adelson, 2013; Sulu-Gambari et al., 2017).

### 3.3.3 Pristane to phytane ratio

The application of pristane to phytane ratio (Pr/Ph) as a redox proxy bases on the theory that both of these aliphatic hydrocarbons are mainly sourced from the phytol side chain of chlorophylls, with preferential diagenetic formation of Ph upon

exposure to reducing conditions (Didyk et al., 1978). Didyk et al. (1978) postulated that Pr/Ph ratios of < 1 are indicative of deposition under anoxic water column, while ratios fluctuating about 1 record oscillations between anoxic and oxic bottom waters, and persistently oxic bottom waters result in ratios > 1. However, it has been shown that a decline in Pr/Ph may also be caused by excess production of Ph sourced from methanogenic microbes below the SMTZ (Brassel et al., 1981; Venkatesan and Kaplan, 1987; Duan, 2000). As a result, the ratio should be used cautiously and in association with other indicators of

redox conditions. This redox proxy is widely used in petroleum geology to characterize source rocks (Peters et al., 2005), but is so far unutilized in studies of Baltic Sea sediments.

## 4 Materials and methods

### 4.1 Sediment coring, subsampling, and analysis of grain size

The study site was selected based on previous studies by Jokinen et al. (2015), where it was found that the sediment in the

basin comprises thick varves since the beginning of the 20[th] century, providing a high-resolution archive of environmental



change for the corresponding period. Due to this apparent sensitivity of the basin to bottom water hypoxia, as manifested in the continuous laminations, together with the central location middle of the Archipelago Sea, we expected the site to be representative of the past environmental changes in the area. Importantly, despite the contrasting bottom water oxygenation between adjacent sub-basins in the area (Virtasalo et al., 2005), the long-term trends in environmental conditions are largely

congruent over the entire Archipelago Sea, excluding areas close to prominent nutrient point sources (Suomela, 2011).

Two replicate sediment cores (HAV-KU-5 and HAV-KU-6) were retrieved using a 5 m long piston corer on board R/V Aurelia of the Archipelago Research Institute in June 2015 (Table 1). The coring device was adjusted to start the core retrieval with the piston positioned 1–2 decimeters above the sediment–water interface, in order to capture the sediment surface as intact as possible. In the laboratory, the cores were split lengthwise and trimmed for digital photography and description of lithology.

All of the following analyses, except for X-radiography, were conducted for the HAV-KU-6 core only. The sediment was sub-sampled into cubic ($7 \text{ cm}^3$) polystyrene sample boxes at approximately every 3.0 cm for geochemical analyses. In addition, the HAV-KU-6 core was continuously sub-sampled at 1 cm resolution into reclosable polyethylene bags for $^{137}\text{Cs}$, $^{14}\text{C}$, and grain size analyses. Grain size distributions were analyzed at every 10 cm by a Coulter LS200 laser diffractometer after pretreatment with excess $H_2O_2$ and dispersing the particles in an ultrasonic bath.

**4.2. Geochronological methods**

Age constraints for the age model were obtained based on visual varve counting (1900 AD onwards), $^{137}\text{Cs}$ profiles, and recognizable features in the measured atmospheric lead (Pb) fallout profile. Three analysts independently counted the varves within the continuously laminated interval (0–76 cm core depth) from the freshly split sediment surface in order to constrain the reproducibility of the method. To fix this floating varve chronology, $^{137}\text{Cs}$ activity was measured for the topmost 60 cm of

the core by measuring the gamma spectra of the wet sub-samples at the Geological Survey of Finland using an EG&E Ortec ACE™-2K spectrometer with a 4 inch NaI/Tl detector. Two or three consecutive 1 cm thick sub-samples were combined for the 20–60 cm and 0–20 cm depth intervals, respectively, to gain enough material for the measurement. No corrections were applied for the results, because the target was only to detect relative $^{137}\text{Cs}$ activity peaks (Jokinen et al., 2015). Due to the lack of datable macrofossils, we also attempted to constrain the age model with bulk sediment radiocarbon dating of the NaOH-

soluble fraction as suggested by Rößler et al. (2011). The NaOH-extraction and AMS-$^{14}\text{C}$ measurements for nine selected core depths were conducted at the Poznań Radiocarbon Laboratory, Poland (Goslar et al., 2004). In addition, we compared our bulk sediment Pb profiles with the Pb fallout records from varved lakes in Sweden (Brännvall et al., 1999) to constrain the chronology.



### 4.3. X-radiography, trace fossils, and bioturbation index

For X-radiography, plastic boxes (50 x 4 x 2 cm) were pressed into the sediment, cut out using a steel string, trimmed with a thin aluminum sheet, and sealed after inserting a lid (Virtasalo et al., 2006). Due to disturbance of the uppermost sediments in the reference half of HAV-KU-6 during splitting of the core, the sub-sampling for X-radiography for the topmost 170 cm was

done for the replicate core HAV-KU-5. For the bottom part, the sub-sampling was conducted for HAV-KU-6 due to the longer core retrieval (Table 1). The cores were correlated visually based on the occurrence of the laminated intervals and the boundaries of the lithological units described below. 2D projections of the sedimentary structures were investigated from high-resolution digital X-radiographs of the boxes, which were obtained by a custom-made tungsten-anode micro-computed-tomography Nanotom device (Phoenix|Xray Systems+ Services GmbH) at the University of Helsinki. Power settings of the

X-ray source were 100 kV and 150μA, and the detector was adjusted to exposure time of 1 s and an averaging of 30 images per radiograph. The resulting pixel size in the X-radiographs was 38 μm. X-radiographs were used for further lithological description as well as for analysis of bioturbation index and ichnological structures to the ichnogenus level (for ichnofossil descriptions, see Virtasalo et al. 2011a) in 6 cm thick sediment intervals. We assigned a bioturbation index of 1−4, modified from Behl and Kennett (1996), to these intervals based on the preservation of the primary sedimentary-fabric so that high

values denote intense mixing (Table 2).

### 4.4 Geochemical analyses

#### 4.4.1 Carbon and nitrogen contents and stable isotope ratios

Lids of the polystyrene sample boxes were removed, the samples were freeze-dried, and the dry bulk density was calculated. Subsequently, the sediment samples were ground in an agate mortar and analyzed for total carbon ($C_{tot}$) and nitrogen ($N_{tot}$) by

CHNS-analyzer (TruSpec Micro, LECO Corporation) in the Department of Food and Environmental Sciences at the University of Helsinki. The amount of inorganic C and N was assumed to be negligible in this setting, and thus $C_{tot}$ and $N_{tot}$ contents are assumed to equal to $C_{org}$ and $N_{org}$, respectively (Jilbert et al., 2017). The relative contributions of terrestrial plant-derived (%$OC_{terr}$) and phytoplankton-derived OM (%$OC_{phyt}$) to the total $C_{org}$ were estimated with a simple two end-member mixing model for the molar N/C ratio, applying end-member values of N/$C_{terr}$ = 0.04 (C/N = 25) and N/$C_{phyt}$ = 0.13 (C/N = 7.7) for

terrestrial and phytoplankton-derived OM, respectively (Goñi et al., 2003):

$$\%OC_{terr} = \frac{(N/C)_{sample} - (N/C)_{phyt}}{(N/C)_{terr} - (N/C)_{phyt}} \times 100 \; , \qquad (2)$$

where %$OC_{terr}$ = 100 - %$OC_{phyt}$. The validity of these end-member values was recently confirmed by a study in the southern coast of Finland, where sediment N/C values for sediment OM covering a transect along the Pojo Bay from the river-mouth to the open Baltic Sea were reported (Jilbert et al., 2017). To obtain the desired fraction of terrestrially derived $C_{org}$ rather than



$N_{org}$ from this mixing model, we use N/C instead of C/N ratio for the calculation (Perdue and Koprivnjak, 2007). Elsewhere in the text and in the figures we refer to the more commonly used C/N ratio.

In the Laboratory of Chronology at the University of Helsinki, selected freeze-dried and ground samples were measured for stable isotope compositions of carbon ($\delta^{13}C_{org}$) and nitrogen ($\delta^{15}N$) by an isotope ratio mass spectrometer (Delta$^{Plus}$ Advantage,
Thermo Fisher Scientific) coupled to an NC2500 elemental analyzer. The measured $\delta^{13}C_{org}$ and $\delta^{15}N$ values are reported relative to the Vienna Pee Dee Belemnite (V-PDB) and AIR scales for C and N, respectively. Precision of the measurements, as checked against in-house and reference standards yielded < 0.02 % (1σ) for both elements. Based on analysis of replicate samples, precision of the entire procedure was < 0.2 % and < 2.6 % for $\delta^{13}C_{org}$ and $\delta^{15}N$, respectively. To account for the historic decline in $\delta^{13}C$ of atmospheric $CO_2$ due to fossil fuel burning and deforestation, the measured values for samples
postdating 1700 AD were corrected according to the equation suggested by Verburg (2007):

$$\%OC_{terr}\delta^{13}C = 7.7738118 \times 10^{-16} \times Y^6 - 1.2222044 \times 10^{-11} \times Y^5 + 7.1612441 \times 10^{-8} \times Y^4 - 2.1017147 \times 10^{-4} \times Y^3 + 3.3316112 \times 10^{-1} \times Y^2 - 273.715025 \times Y + 91703.261 , \qquad (3)$$

where Y = year (AD) of the sediment accumulation.

### 4.4.2 Biomarker analyses

Selected freeze-dried and homogenized samples were analyzed for various biomarkers at the Department of Marine Geology in the Leibniz Institute for Baltic Sea Research (IOW) following Kaiser and Arz (2016). Briefly, 0.5−1.0 g of sediment was used for accelerated solvent extraction (Dionex ASE 350, Thermo Fisher Scientific) with a 9:1 volumetric mixture of dichloromethane and methanol using high pressure (100 bar) and temperature (100 ⁰C). After the addition of internal standards (squalane, nonadecan-2-one, 5α-androstan-3β-ol, and $C_{46}$-GDGT) for quantification, the total extracts were divided into four
fractions by microscale silica gel column chromatography. For the determination of Pr and Ph contents, the apolar alkane fraction was measured on a multichannel Trace-Ultra Gas Chromatograph (Thermo Fisher Scientific), utilizing split/splitless inlet, a DB-5 MS capillary column and a FID detector. Peak identification from the obtained chromatograms was done based on comparison of peak retention times with an external standard containing Pr, Ph and $n$-$C_8$ to $n$-$C_{40}$ alkanes, complemented with GC-MS analyses (see for details Kaiser and Arz, 2016). To calculate the BIT index, the most polar fraction (including
glycerol dialkyl glycerol tetraethers, GDGTs) was measured with HPLC APCI-MS (Dionex Ultimate 3000 UHPLC, Thermo Fisher Scientific system coupled to a MSQ Plus, Thermo Fisher Scientific) (see for details Kaiser and Arz, 2016).

### 4.4.3 Major and trace element contents

Elemental contents of the freeze-dried and homogenized samples were estimated by a combination of ICP-OES and ICP-MS analysis. Initial ICP-OES analysis for K and Ti was performed at the Department of Food and Environmental Sciences at the



University of Helsinki. 0.1−0.2 g of dry sediment was dissolved in 2.5 mL of HF (38 %) and 2.5 mL of a mixture of $HClO_4$ (70 %) and $HNO_3$ (65 %) (volumetric ratio 3:2) in closed Teflon bombs at 90 ℃ for 12 h. After evaporating the acids at 160 ℃, the remaining gel was dissolved in Suprapur[R] 1 M $HNO_3$ and analyzed for K and Ti by ICP-OES (Thermo Fisher Scientific, precision determined by replicate analyses < 5 %).

Accuracy of the initial ICP-OES results was checked by digestion and analysis of a subset of 28 samples at IOW together with the international reference material SGR-1b (USGS). 50 mg of dried and ground sediment was first treated in open Teflon vessels (PDS-6; Heinrichs et al., 1986) with 1 mL $HNO_3$ (65 %) at 60 °C for 1 h to oxidize OM. After addition of 2 mL concentrated HF and 2 mL concentrated $HClO_4$, the closed vessels were heated at 180 ℃ for 12 h. After evaporation of the acids on a hot plate at 180 °C, the digestions were fumed off 3-times with 6 M HCl, re-dissolved in 25 ml 2 vol % $HNO_3$ and

finally measured by ICP-OES (iCAP 7300 Duo, Thermo Fisher Scientific) for K and Ti using Sc as internal standard. Precision and accuracy of the measurements of SGR-1b at IOW were 0.7 %/-5.9 % for K and 0.8 %/-7 % for Ti. Constant offsets of up to 20% were observed between the datasets from Helsinki and IOW for the 28 samples. A correction factor was thus applied to the results from the Helsinki data using a linear regression between the IOW and Helsinki results. The K and Ti data reported in this manuscript are thus the corrected Helsinki data, and may be considered to have precision and accuracy < 10 % (this

value integrates both the quality of the IOW measurements and the goodness of fit of the linear regression). Internal reproducibility between the two sample sets was good (< 7 %), suggesting that the offsets observed between the respective ICP-OES datasets from Helsinki and IOW were related to analytical issues rather than the digestion protocols.

ICP-MS was used to determine the contents of Mo, Pb and the ratio of stable Pb isotopes ($^{206}Pb/^{207}Pb$). ICP-MS analysis (iCAP Q, Thermo Fisher Scientific) was performed at IOW on the complete set of digests from Helsinki, and the subset of 28 digested

samples from IOW. The international reference material SGR-1b (USGS) served to determine precision and accuracy of Mo (0.7 %/-1.6 %) and Pb (0.6 %/-2.2 %) measurements using Rh and Ir as internal standards in KED mode using He as collision gas. For determination of $^{206}Pb/^{207}Pb$ in standard mode, the samples were diluted to a Pb concentration of ~ 1 µg $L^{-1}$ and the instrument was tuned to provide best results for the NIST standard SRM-981 resulting in a precision < 0.07 %.

All measured elemental contents were corrected for the weight of the salt in the pore water using the ambient salinity and

porosity (Lenz et al., 2015). Mass accumulation rates (MARs) of the individual elements were calculated as follows:

$$MAR_x = \frac{C_x}{100} \times LSR \times DBD \; , \qquad\qquad (4)$$

where $C_x$ = element content (%), LSR = linear sedimentation rate (cm $a^{-1}$), DBD = dry bulk density (g $cm^{-3}$), and subscript x denotes the element in question.



## 5 Results

### 5.1. Age model

The three independent varve counts for the continuously laminated recent sediments (3-76 cm) suggest that this section covers
$113 \pm 5$ varve years of deposition. The $^{137}$Cs activity increase at 33 cm core depth, derived from the Chernobyl nuclear power

plant accident in 1986 AD, enabled us to fix the floating varve chronology, which indicated that the onset of continuous
lamination occurred at ~ 1900 AD (Fig. 2). A similar $^{137}$Cs activity increase at this site was found at 32 cm core depth in a core
taken in 2013 AD (Jokinen et al., 2015), implying that no marked loss of surface sediment occurred during the piston coring.
This is supported by the varve counting, which (after fixing with the $^{137}$Cs activity profile) suggested that the sediment surface
was indeed of modern age (Table 3).

The apparent temporal fluctuations in the magnitude of the bulk sediment reservoir age as indicated by the downcore reversals
in $^{14}$C age (Fig. 2; Lougheed et al., 2017) impeded utilization of the bulk sediment AMS-$^{14}$C dates for the construction of the
age model. Because our data did not enable robust correction for such variations, we desisted from using bulk sediment $^{14}$C
dates as age constraints.

To estimate the content of atmospheric pollution Pb in the sediment, we applied a simple two-component mixing model

following Brännvall et al. (1999):

$$\text{Pb}_{pollution} = \frac{(^{206}\text{Pb}/^{207}\text{Pb})_{sample} - (^{206}\text{Pb}/^{207}\text{Pb})_{background}}{(^{206}\text{Pb}/^{207}\text{Pb})_{pollution} - (^{206}\text{Pb}/^{207}\text{Pb})_{background}} \times \text{Pb}_{sample} \tag{5}$$

The background isotope composition of Pb was calculated as the mean prior to the apparent onset of Pb pollution at 900 AD
(Fig. 2), which yielded an estimate of 1.397. We used the age model based on $^{137}$Cs dating and varve counting for the sediments
deposited after 1900 AD, to assign a time-dependent $^{206}$Pb/$^{207}$Pb composition for the pollution Pb. We assumed a constant

$^{206}$Pb/$^{207}$Pb ratio of 1.17 for the pollution Pb prior to 1900 AD, a linear decrease from 1.17 to 1.15 between 1900 and 1945 AD,
and a constant ratio of 1.15 since 1945 AD (Brännvall et al., 1999 and references therein).

We found a remarkable consistency between our Pb record and the pollution Pb profiles of varved lakes in eastern Sweden,
where the pre-industrial Pb fallout was mainly sourced from mainland Europe and from the British Isles (Brännvall et al.,
1999; Fig. 2). Hence, we used the main Pb pollution features found consistently in all of these lakes to constrain an age model

for our sediment core (Fig. 2; Table 3). The onset of Medieval Pb pollution (at 900 AD, Lougheed et al., 2012) as well as the
Medieval pollution maximum (at 1200 AD, Lougheed et al., 2012; Zillén et al., 2012; Lougheed et al., 2017) have previously
been used as age constraints in sediments from the Baltic Proper. However, our study is, to our best knowledge, the first to



identify the 1530 AD pollution peak and the pollution minima of 1350 and 1600 AD (Brännvall et al., 1999; Renberg et al., 2002) as age–depth points in Baltic Sea sediments.

A conservative 1σ uncertainty of 50 years for all of the Pb constraints was assumed. On top of this, we assigned an additional uncertainty for the Pb age constraints in cases where the position of the feature was ambiguous by first calculating the mean

LSR for the 900−1900 AD period (from the sharp onset of Pb pollution to the onset of continuous lamination). Based on this LSR estimate (0.21 cm a$^{-1}$) and the width of the Pb features, an additional uncertainty was added to these constraints by assuming that the width of each recognized feature in our Pb profile corresponds to the 2σ uncertainty in the correlation with the Swedish pollution Pb records.

The obtained age constraints (Table 3) were used to construct a Bayesian age model in OxCal 4.2 software (Bronk Ramsey,

2009), using P-sequence function (Bronk Ramsey, 2008) with a k-value of 20. The resulting age model indicates that the core HAV-KU-6 covers ~ 1500 years of depositional history in the area (Fig. 2). A substantial increase in LSR is observed coincident with the onset of laminated sediment accumulation at 76 cm depth (Fig. 2).

## 5.2 Lithology and trace fossils

The sediment in the core HAV-KU-6 is characterized by organic-rich mud, which conforms to the brackish-water mud drift

sensu Virtasalo et al. (2007), indicative of recent deposition in the basin. Based on lithology, the core can be divided into four units as described below (Figs. 2 and 3). These units approximately correspond to the pre-MCA, MCA, LIA and MoWP intervals (Fig. 3).

The basal part (393–324 cm, ~ 500–780 AD) of the core HAV-KU-6 roughly corresponds to the pre-MCA interval, and comprises seemingly homogenous greenish-brown mud. The X-radiographs show that this mud is intensely burrow-mottled

(*Planolites*-mottling) and is thus characterized by a bioturbation index of 4 (Fig. 4), although discrete trace fossils are hardly discernible due to the high burrow density and low contrast enhancement by pyritization along the burrows (Fig. 3a), probably resulting in substantial underestimation of the number of traces. The upper contact to the thinly-bedded mud is relatively sharp, but no clear signs of erosion are observed in the X-radiograph.

The thinly-bedded mud (324–201 cm, ~ 780–1290 AD) roughly corresponds to the MCA interval, and comprises 2–5 cm thick

beds, which are burrow-mottled and poorly graded, and locally inverse graded (Fig. 3b). Bioturbation index in this unit varies from 3 to 4 depending on the preservation of the beds (Fig. 4). In places where the bedding pattern is clearly visible, the basal parts of the beds are dark grey and appear dark in the X-radiographs, suggesting relatively low density. In contrast, the upper parts of the beds are greenish-grey and appear bright in the X-radiographs, pointing to higher density and coarser grain-size.



The beds are mottled by abundant *Planolites*, rare to abundant *Arenicolites*, and rare large *Planolites* ichnofossils and complex bivalve biodeformational structures (Figs. 3b and 4).

The thinly-bedded mud is gradationally overlain by greenish-grey mud roughly corresponding to the LIA interval (201–76 cm, ~ 1290–1900 AD), with occasional 1–4 cm thick indistinctly laminated intervals. In general, this unit is characterized by a

complete obliteration of the primary sedimentary-fabric (bioturbation index of 4) and the trace fossil assemblage comprises abundant *Planolites* and *Arenicolites*, rare large *Planolites*, and complex bivalve biodeformational structures (Figs. 3c and 4). Bioturbation index and the number of traces decline rapidly at 91 cm core depth (~ 1820 AD), and remain low to the top of the unit (Fig. 4). The upper contact to the sharply laminated mud is gradational.

The sharply laminated mud (76–0 cm) roughly corresponds to the MoWP and comprises rhythmically alternating light brown,

black, and grey laminae. The thickness of the individual lamina successions varies from 2 to 13 mm, with a generally upward increasing trend, which partly results from decreasing compaction. These laminites correspond to the varves described by Jokinen et al. (2015). Trace fossils are virtually absent in this unit (bioturbation index of 1–2), although occasional blurring of the laminations is observed (Figs. 3d and 4). Black sulfide-staining becomes predominant from 53 cm upwards.

### 5.3 Geochemistry

### 5.3.1 Lithogenic components: trends over the entire study interval

Median grain-size (range 1.8–2.4 µm) and Ti/K (range 0.146–0.159) are closely coupled, and show a generally decreasing trend towards the present, (Fig. 4). Superimposed on this trend the LIA stands out as an interval of decreased values in both profiles (median grain size ~ 1.9 µm, Ti/K ~ 0.149) in comparison to the pre-MCA, MCA and MoWP.

### 5.3.2 Organic component: trends over the entire study interval

The proxies for organic matter contents and composition correlate strongly with each other and show profiles very similar to those in Ti/K and grain size (e.g., Ti/K vs. $C_{org}$ $r_s = 0.83$, $p < 0.001$; Table 4). Namely, the organic proxies display a long-term trend (towards lower $C_{org}$ content and BIT, and higher C/N) onto which changes during the MCA and MoWP (towards higher $C_{org}$ content and BIT, and lower C/N) are superimposed (Fig. 5). Despite the relatively narrow ranges of values for $C_{org}$ (2.9–4.5 %), C/N (8.6–9.9), BIT index (0.18–0.29) and $\delta^{13}C_{org}$ (-23.9–-22.2 ‰), variability in these proxies are internally consistent

over the last 1500 years (Fig. 5). The MAR of $C_{org}$ remained between 20 and 50 g m$^{-2}$ a$^{-1}$ until the onset of the 20th century, after which it increased up to ~ 100 g m$^{-2}$ a$^{-1}$ by the end of the century. Meanwhile, the bulk sedimentary $\delta^{15}N$ remained fairly constant (3.1–3.5 ‰) throughout much of the study period, until a steady increase from 3.4 to 5.2 ‰ in the 1900s.



### 5.3.3 Proxies for hypoxia: trends over the entire study interval

Sedimentary Mo content and MAR (ranges 2–8 mg kg$^{-1}$ and 0.09–1.21 µg cm$^{-2}$ a$^{-1}$) display negligible changes throughout the pre-MCA, MCA and LIA intervals, but increase to maximum values during the MoWP (Fig. 4). When expressed as MAR, the recent increase in Mo continues to the present day (Fig. 6). The correlation between Mo and C$_{org}$ is generally weak (r$_s$ = 0.18,

p = 0.04) although this improves towards the present (i.e. when considering the LIA and MoWP intervals only, Fig. S1).

Pr/Ph ratio also shows little systematic change throughout most of the record, recording values in the range of 0.4 – 1.0 from the pre-MCA to the early MoWP. Coincident with the sub-recent increase in Mo, Pr/Ph drops sharply to nearly zero at the depth interval corresponding to 1960 AD, before recovering to 0.6 in the most recent sediments (Fig. 4). Throughout the entire record, Mo and Pr/Ph show a negative correlation (r$_s$ = -0.48, p < 0.001; Fig. S1; Table 4).

### 5.3.4 Trends during the Modern Warm Period

The recent changes observed in our proxies during the MoWP are complex, and are described here in more detail. The LIA– MoWP transition (1800–1900 AD) was characterized by low Mo and C$_{org}$ contents and MARs (range during the MoWP 0.17– 1.21 µg cm$^{-2}$ a$^{-1}$ and 22–122 g m$^{-2}$ a$^{-1}$, respectively) as well as low sediment MAR (range during the MoWP 0.7–2.9 kg m$^{-2}$ a$^{-1}$) and high Pr/Ph (Fig. 6). At 1900 AD, coinciding with the onset of continuous laminations, MARs of Mo, C$_{org}$, and sediment

increased contemporaneously with a marked decrease in Pr/Ph. After this point, the strongly enhanced sediment MAR appears to partially dilute the Mo content, particularly in the late 20$^{th}$ century (see also Fig. 4). Hence we focus on the MAR of Mo as a proxy for recent changes in hypoxia during the MoWP. The Mo MAR shows a steady increase throughout the MoWP, displaying several decadal-scale oscillations (Fig. 6).

Despite declining rapidly after the onset of continuous laminations, Pr/Ph recovered to pre-MoWP values in the period 1990

AD–present (Fig. 6). The possible reasons for this observation are discussed below. Meanwhile, the proxies for organic matter composition during the MoWP are consistent with a steady increase in the relative supply of autochthonous material. δ$^{13}$C$_{org}$ values increase progressively from 1900 AD to present (-23.6–-22.2 ‰), while C/N declines from 9.8 to 9.1.

## 6 Discussion

### 6.1 Physical changes in the depositional conditions

Although the Haverö basin has been an enclosed basin throughout the study period (Fig. 1b), the glacio-isostatic rebound has resulted in progressively calmer depositional conditions in the area, as evidenced by the generally decreasing trajectories in Ti/K and grain-size over the past 1500 years (Fig. 4). Assuming a constant glacio-isostatic uplift rate of 4 mm a$^{-1}$ for the past 1000 years (Mäkinen and Saaranen 1998), and taking into account the ~ 3 m of sediment accumulated during this period, the




basin was ~ 7 m deeper around 1000 AD than at present. In addition, the intensive sediment focusing and drift-like deposition of the brackish-water muds has likely smoothed the bottom topography (Virtasalo et al., 2007), which has further decreased the bottom water volume towards present times. The gradual shoaling of the basin has decreased the area located below the wave-base, which in combination with the steep shoreline topography has increased the sediment source-to-sink ratio and led to more effective sediment focusing in the deepest parts of the basin towards the present, as implied by the enhanced sediment MAR during the MoWP (Fig. 6).

Superimposed on this general trend towards calmer sedimentary conditions towards the present, climatic oscillations have exerted a prominent control on the bottom water hydrographic conditions at the study site. We ascribe the elevated Ti/K and slightly coarser grain size during the MCA and MoWP in comparison to the LIA to enhanced bottom water currents and lateral sediment transport (Fig. 4; Spoffort et al., 2008), whereby the lack of ice cover in the late autumn–early winter promoted wave-induced sediment focusing upon exposure to storms. Conversely, the early formation of ice cover during the LIA likely suppressed wind-induced mixing of the water column (Lincoln et al., 2016), thereby reducing bottom water energy flux. Accordingly, it has been shown that a major proportion of the annual sediment accumulation in the Haverö occurs in late autumn and early winter due to the interplay of intensified cyclonic activity and lack of ice cover (Jokinen et al., 2015).

## 6.2. Organic matter input

### 6.2.1 Source of the deposited organic matter

The bulk sediment $\delta^{13}C_{org}$ signature (from -23.9 to -22.2 ‰) together with the low C/N ratios (from 8.6 to 9.9) indicates that most of the OM deposited in the Haverö basin originates from autochthonous primary productivity rather than from terrestrial sources (e.g. Meyers, 1994; 1997), agreeing with the findings from analogous settings in the Swedish archipelago areas (Jönsson et al., 2005b; Savage et al., 2010). The marked positive correlation of BIT index and C/N ratio, and their negative correlation with $\delta^{13}C_{org}$ (Fig. 5; Table 4) implies that all of these parameters are sensitive to variations in the source of OM. The two-component mixing model for N/C in the sediment OM suggests that, on average, 23 % (range from 15 to 32 %) of the sediment $C_{org}$ originates from terrestrial sources, the highest contribution being reached when the amount of phytoplankton-derived $C_{org}$ is at minimum (Fig. 5). Similarly, conversion of the BIT index values to percentages of terrestrial (soil) OM (%$OM_{terr}$ = 100 * BIT) yields an average estimate of 22 % (range from 18 to 29 %). These figures are in line with the N/C-based estimates of ~ 20 % for %$OC_{terr}$ in the middle parts of the archipelago area south of the Pojo Bay, in the Gulf of Finland (Jilbert et al., 2017). Collectively, these observations demonstrate that the changes in OM deposition are mainly controlled by variations in the autochthonous, phytoplankton-derived input.

We note that the excessively old bulk sediment radiocarbon dates for the NaOH-soluble fraction suggest a marked input of old reworked OM (Fig. 2). Despite this contribution of pre-aged carbon, likely related to intensive reworking and lateral sediment





advection in the basin (Jokinen et al., 2015), we assume that the $C_{org}$ content and $\delta^{13}C_{org}$ signature of the redeposited material have remained relatively constant, because this material represents a spatial and temporal mixture of sediments deposited in the area (Struck et al., 2000). The enclosed configuration of the basin throughout the study interval suggests that the majority of OM that has ultimately settled at our study site likely originates from a rather small area, and thus the observed increases in

OM accumulation are likely caused by local enhanced productivity. Nevertheless, the intensive reworking and lateral sediment transport is likely to dilute and smooth the signal of the delivery of OM sourced from the contemporaneous local primary productivity in the euphotic zone, partly explaining the generally subtle variability in the sediment C/N and $\delta^{13}C_{org}$ profiles.

### 6.2.2 Temporal fluctuations in the organic matter input

The fluctuations in the input of phytoplankton-derived OM to the basin generally coincide with the past climatic oscillations

(Fig. 5) and with paleoproductivity records from the Baltic Proper (Leipe et al., 2008; Kabel et al., 2012; Jilbert and Slomp, 2013). Indeed, despite the negative long-term trend in autochthonous OM concentration from the pre-MCA towards the present, the MCA and MoWP are typified by relatively high input in comparison to the LIA, implying enhanced productivity under warm climatic phases. However, we note that $\delta^{13}C_{org}$ and the MAR of $C_{org}$ during the MCA remained below the MoWP values, indicating that the export production of OM and consequent sediment focusing never reached the MoWP levels. By

contrast, a distinct decrease in $C_{org}$ and $\delta^{13}C_{org}$, paralleled by an increase in C/N and BIT index, coincides with the MCA–LIA transition (Fig. 5), suggesting that this marked decline in local primary productivity was most likely forced by the climatic cooling (Kabel et al., 2012).

The temporal trend in OM input observed in our data is similar to the trend in temperature recorded in millennial-scale regional climate model simulations (Schimanke et al., 2012), and with dendroclimatic summer temperature reconstructions for south-

eastern Finland (Helama et al., 2014) as well as with $TEX_{86}$-derived sea surface temperature reconstruction for the Baltic Proper (Kabel et al., 2012). All of these studies report a decline in temperatures at ~ 1300–1400 AD with a persistently cold period lasting until ~ 1900 AD, similarly to the fluctuations in the OM input at our study site (Fig. 5). Furthermore, the period of lowest productivity at ~ 1700–1900 AD, as indicated by the maximum values in C/N and BIT and concurrent minimum in the $\delta^{13}C_{org}$ signature, is sympathetic with a minimum in summer temperatures in south-eastern Finland (Fig. 5; Helama et al.,

25    2014).

Importantly, we note that the long-term trends in land-use and precipitation in the catchment contradict with our record of phytoplankton-derived OM input prior to the MoWP, suggesting that external nutrient inputs did not force past increases in productivity. Indeed, in agreement with the population growth records for Finland (Kuosmanen et al., 2016 and references therein), marked human-induced land-use changes in the catchments of varved lakes in south-western and south-central

Finland became notable at ~ 1400 AD (Tiljander et al., 2003; Ojala and Alenius, 2005). This interval coincides with the onset



of the LIA, during which both precipitation (Väliranta et al., 2007; Helama et al., 2009; Luoto, 2009; Saarni et al., 2015) and soil erosion rate (Tiljander et al., 2003; Ojala and Alenius, 2005; Saarni et al., 2016) increased in south-central Finland. However, this period is characterized by a shift to suppressed primary productivity at our study site, implying no influence of enhanced external nutrient inputs. The lack of anthropogenic forcing of OM input during the MCA is also evidenced by the

constant sediment $\delta^{15}N$ signature over this period at Haverö (Fig. 5; see also Cole et al., 2004). Finally, indirect anthropogenic influence via nutrient transport from the Baltic Proper, claimed to have received considerable human-induced nutrient loading already during the MCA (Zillén and Conley, 2010), is unlikely as the nutrient input in the study area is mainly driven by the local nutrient sources (Hänninen et al., 2000) due to the restricted water exchange with the open sea areas (Mälkki et al., 1979).

The progressive increase in the $\delta^{15}N$ values, beginning already before the turn of the 20th century, attests to recent increased
anthropogenic nutrient loading from agriculture and urban sources to the Baltic Sea (Voss and Struck; 1997; Struck et al., 2000; Voss et al., 2000; Savage et al., 2010). This is supported by the contemporaneous exponential population increase in Turku, which combined with the construction of a sewer network for the city likely enhanced sewage loading to the Archipelago Sea (Fig. 6). Yet, we observe negligible changes in the source of OM around 1900 AD, as implied by the relatively constant C/N and $\delta^{13}C$ values (Fig. 6). This suggests that the strong increase in the $C_{org}$ MAR at this time was mainly caused
by intensified sediment focusing, as supported by the contemporaneous increases in Ti/K and sediment MAR (Figs. 4 and 6). We propose that the general trend towards higher source-to-sink ratio in the basin, combined with the climate-driven intensification of wind-induced sediment reworking (Fig. 6) to increase the $C_{org}$ MAR to the sediments. By contrast, the shift to unprecedentedly heavy $\delta^{13}C$ signature, sympathetic with a decrease in C/N and BIT index around 1950 AD (Fig. 6), points to direct enhanced export production of phytoplankton-derived OM at this time (Meyers, 1994; 1997; Hopmans et al., 2004).
This later shift was likely fueled by the concomitant increase in nutrient loading from urban and agricultural sources in the catchment (Fig. 6), being in line with eutrophication trajectories elsewhere in the Baltic Sea linked to amplified anthropogenic nutrient inputs (Struck et al., 2000; Conley et al., 2009a; Savage et al., 2010; Gustafsson et al., 2012; Carstensen et al., 2014a).

### 6.3 Changes in hypoxia intensity and its causes

### 6.3.1 Bottom water oxygenation prior to the Modern Warm Period

The intensive *Planolites*-mottling that completely overprints the primary sedimentary-fabric (bioturbation index 4) in the pre-MCA sediments clearly demonstrates efficient ventilation of the bottom waters, which enabled macrobenthic fauna to inhabit the seafloor (Figs. 3a and 4). The scarcity of discrete traces in this sediment interval could be attributed to less intense pyritization along the burrow walls than in the overlying sediments (Thomsen and Vorren, 1984) and to the overlapping of densely spaced burrows (Virtasalo et al., 2011a, b). Macrofauna living in the area and capable of completely mixing the
seafloor sediment include the isopod *Saduria entomon,* the polychaete *Harmothoe sarsi*, the amphipods *Monoporeia affinis*





and *Pontoporeia femorata*, and the mysid shrimp *Mysis relicta*. The discrete *Planolites* are produced by vermiform burrowers such as the priapulid *Halicryptus spinulosus*.

The enhanced preservation of the thinly-bedded sedimentary-fabric during the MCA at ~ 900–1200 AD, as demonstrated by the decrease in the bioturbation index (Figs. 3b and 4), is indicative of declining near-bottom oxygen levels during this period.

The poorly developed grading of these beds is ascribed to the predominantly lateral sediment transport in the basin (Jokinen et al., 2015), and possibly results from the slow migration of thin accretionary bedforms. The dominance of *Planolites* with a small diameter and vertical extent in the trace fossil assemblage together with the subtle decrease in the bioturbation index (Fig. 4) demonstrate low burrowing activity under a considerable oxygen stress (Savrda and Bottjer, 1986, 1991), especially during the first half of the MCA.

Despite the decline in the bottom water oxygen concentration during the MCA, we infer that the pore water chemistry was typified by a relatively deep and poorly-developed SMTZ with low $H_2S$ concentration, which hampered efficient Mo sequestration (Helz and Adelson, 2013) and Ph production by methanogenic microbes (Sect. 6.3.2). This could explain why the Mo and Pr/Ph profiles fail to record the modest deoxygenation during the MCA indicated by the trace fossil data. Since ~ 1200 AD, the increases in the bioturbation index and in the abundance of *Arenicolites* trace fossils, characterized by generally

larger size and greater vertical extent than *Planolites*, denote enhanced bottom water oxygenation and downward shift of the SMTZ, continuing throughout the LIA (Figs 3c and 4). Indeed, the greater burrow depth and size suggests downward shift in the annual mean depth of the redoxcline during this period (Savrda and Bottjer, 1986, 1991).

We attribute the multicentennial-scale fluctuations in bottom water oxygenation associated with the MCA and LIA to climatic variability that modulated both hydrographic conditions and accumulation of OM at the sea floor. This is supported by the

Ti/K and grain size profiles and the organic matter proxies (Sects. 6.1 and 6.2) that indicate amplified lateral sediment transport (focusing) and primary productivity during the MCA in comparison to the LIA (Figs. 4 and 5). In addition, we note a slight decrease in LSR at ~ 1380 AD (Fig. 2), coinciding with the MCA–LIA transition. The temporal pattern is similar to the development of hypoxia in the Baltic Proper (Jilbert and Slomp, 2013; Funkey et al., 2014; Jilbert et al., 2015; Dijsktra et al., 2016; Hardisty et al., 2017) and in the Danish Straits (van Helmond et al., 2017), where warm climatic phases during the

Holocene have been characterized by declining oxygen levels. Yet, we note that amplitude of these changes appears markedly less pronounced at our study site, which is most likely attributed to the lack of permanent halocline in this shallow coastal region. Collectively, the similar trends in the intensity of hypoxia in different parts of the Baltic Sea basin attest to regional forcing that was most likely related to changes in the atmospheric circulation patterns such as the NAO, as suggested by Jilbert and Slomp (2013).



### 6.3.2 Progressive intensification of hypoxia during the Modern Warm Period

The MAR of Mo in the sediments at our study site increased during the MoWP (Fig. 6). Due to the fact that the study site is seasonally hypoxic, rather than permanently anoxic or euxinic, the most likely mechanism for Mo enrichment is via diffusion of seawater Mo into the sediment towards the SMTZ (see Helz and Adelson, 2013), which may be amplified by the shuttling

of Mo associated with Mn oxides (Algeo and Lyons, 2006; Scheiderich et al., 2010; Scott and Lyons, 2012; Sulu-Gambari et al. 2017). It is thus plausible that the scavenging of Mo mostly takes place close to the sediment surface, at the end of summer stratification period when the SMTZ reaches its shallowest position in the sediment column (Mogollón et al., 2011) and shuttling and refluxing of Mn is expected to be at its annual maximum. Indeed, total sulfide concentrations of $> 100$ µM have been reported to prevail in the uppermost centimeters of the sediment column within the SMTZ in an analogous setting in the

coastal Baltic Sea (Jilbert et al., 2017). Assuming a salinity and temperature of 5.6 and 9 °C, respectively (Virtasalo et al., 2005) and a pH of 6.6 (our unpublished data) at the sediment−water interface, this total sulfide concentration corresponds to $H_2S_{aq}$ concentration of $> 70$ µM (calculated with R package AquaEnv, Hoffmann et al., 2010), clearly exceeding the requirement of 11 µM $H_2S_{aq}$ for the activation of the sulfide-switch (Helz, et al., 1996; Erickson and Helz, 2000). The annual accumulation rate of Mo is expected to be controlled by the duration of the late summer period when the SMTZ is located

closest to the sediment surface (Adelson et al., 2001; Helz and Adelson, 2013). By extension, when this period is longer in duration we expect seasonal hypoxia in the bottom water to be more intense, and thus that the Mo accumulation rate provides a first-order proxy for bottom water hypoxia during the MoWP. We note that the amount of anthropogenically-sourced Mo in our sediment record is likely negligible in comparison to the enrichment caused by authigenic processes. Indeed, it has been shown that modern sediment sequestration of Mo in the area shows no spatial trends, but is largely controlled by bottom water

oxygenation (Peltola et al., 2011).

During the MoWP, the Haverö basin has undergone a progressive aggravation of bottom water hypoxia, typified by two distinct regime shifts. First, a marked shoaling of the sediment redoxcline at 1900 AD is manifested in the contemporaneous occurrence of continuous lamination (near-complete cessation of macrobenthic activity), a decrease in Pr/Ph, and an increase in the MAR of Mo (Fig. 6). Although the onset of the increased Mo MAR is hard to constrain due to the scarcity of age constraints prior

to the preservation of continuous laminations, the appearance of subtle enrichments in the Mo content evidence intermittent shoaling of the SMTZ and consequently intensified sequestration of Mo in the sediments since 1900 AD (Adelson et al., 2001; Helz and Adelson, 2013). Likewise, the preservation of laminated sedimentary-fabric suggests upward migration of the redoxcline towards the sediment-water interface, inhibiting burrowing by macroinfauna, whereas the occasional blurring of laminations (Fig. 3d) is ascribed to subtle mixing by meiofauna or transient colonization by opportunistic nectobenthos

(Virtasalo et al., 2011b; Jokinen et al., 2015). Although annual recovery of macrofauna is common to systems prone to recurring seasonal hypoxia (Diaz and Rosenberg, 1995; Levin et al., 2009), such rapid recolonization was effectively inhibited in the Haverö basin since 1900 AD, possibly owing to increased porewater $H_2S$ concentration (Diaz and Rosenberg, 1995). A



critical threshold for defaunation in areas experiencing seasonal hypoxia is often around 0.7 mg L$^{-1}$ (Llansó, 1992; Diaz and Rosenberg, 1995), pointing to severe near-bottom water oxygen depletion at the study site already at 1900 AD.

Considering the negligible variation in the proxies for the source of OM prior to 1930 AD (Fig. 6), the onset of seasonal hypoxia and the resulting preservation of continuous lamination since the beginning of 20th century was apparently not forced by changes in primary productivity. Instead, we postulate that this deoxygenation was forced by the following complex interplay of warming climate and millennial-scale changes in the basin configuration: (1) increased source-to-sink ratio, combined with intensified lateral sediment transport especially during early winter, enhanced sediment focusing and the MAR of $C_{org}$ (Sect. 6.2.2); (2) decreased bottom water volume due to the gradual shoaling of the basin resulted in increased vulnerability to hypoxia (Caballero-Alfonso et al., 2015); (3) strengthened summer thermocline caused by global warming and gradual isolation of the basin hampered replenishment of the bottom water oxygen reservoir (Hordoir et al., 2017); (4) continuous accumulation of organic-rich brackish-water muds in the basin at least since ~ 500 AD has resulted in progressive depletion of electron acceptors ("oxygen debt") at the sea floor (Pamatmat, 1971). Accordingly, although we can not completely exclude the possible contribution of anthropogenic forcing, the onset of recurring seasonal hypoxia at around 1900 AD can be largely attributed to the naturally increased vulnerability to deoxygenation that together with global warming irreversibly tipped the ecosystem over a threshold, inducing a regime shift commonly associated with coastal oxygen deficiency (e.g. Conley et al., 2009b).

Another marked redox shift is observed at 1950 AD, where a rapid increase in Mo MAR accompanied by a prominent decrease in Pr/Ph suggest unprecedentedly reducing conditions at the sediment–water interface (Fig. 6). This shift likely denotes shoaling of the redox zonation as a response to eutrophication in the area, which has been reported in previous studies of the Baltic Sea (Slomp et al., 2013; Egger et al., 2015; Rooze et al., 2016: Jilbert et al., 2017) and reflects intensified delivery of labile OM to the seafloor with respect to the supply of electron acceptors (Middelburg and Levin, 2009). Considering the decreasing trend in the local summer temperatures over this interval, possibly promoting weaker thermal stratification and suppressed productivity, we infer that the upward migration of the SMTZ was more likely driven by the increased anthropogenic nutrient loading from the catchment than by climatic factors (Fig. 6). Similar exacerbation of bottom water hypoxia around 1950s has been reported in the coastal areas of Sweden (Persson and Jonsson, 2000; Savage et al., 2010) and in the Baltic Proper (Fig. 6; Jonsson et al., 1990; Carstensen et al., 2014a), reflecting synchronous increases in the anthropogenic nutrient loading around the Baltic Sea (Conley et al., 2009a; Savage et al., 2010; Gustafsson et al., 2012 Carstensen et al., 2014a). This deoxygenation around 1950s conforms to the global spread of hypoxia in coastal areas (Vaquer-Sunyer and Duarte, 2008) following the "Great Acceleration" (Steffen et al., 2015). Accordingly, the steepest gradient in the $\delta^{15}N$ profile at our study site is reached between 1950 and 1995 AD, paralleled by intensified agricultural practices and fast population growth rate (Fig. 6).





Although the decline in Mo content and concomitant increase in Pr/Ph suggest a slight improvement in the bottom water oxygenation since 1990 AD (Fig. 6), this is an unlikely scenario. Instead, the MAR of Mo has remained high since 1950 AD and reached the highest values of the record as late as 2005 AD, suggesting that aggravation of bottom water hypoxia has continued up to the present, as evidenced by the local monitoring data directly at the study site (Fig. S2). In line with this,

monitoring data in other parts of the Archipelago Sea consistently demonstrate progressive deoxygenation in the area during last two decades (Fig. 7; Suomela 2011; Caballero-Alfonso et al., 2015). Therefore, we ascribe the decrease in Mo content at 1990 AD to dilution by the concurrently enhanced sediment MAR (Fig. 6). We postulate that, in addition to effective sediment focusing, the increased sediment MAR was likely fueled by ballasting effects, whereby eutrophication-induced increase in the OM production in the euphotic zone drives the sedimentation of fine-grained lithogenic material through aggregation (Passow,

2004; Passow and De La Rocha, 2006; De La Rocha et al., 2008), as suggested by the close covariation between sediment and $C_{org}$ MARs (Fig. 6). Indeed, it has been shown that rapid sedimentation events during vernal phytoplankton blooms in the study area are caused by the formation of organomineralic aggregates adhered together by sticky transparent exopolymers (TEP) excreted by phytoplankton (Jokinen et al., 2015). Therefore, while $C_{org}$ content remained fairly constant over the shift to enhanced sediment MAR at 1985 AD due to intensified sediment focusing and ballasting effects, the content of Mo was

diluted as the depth and intensity of the $H_2S$ front remained relatively unaffected.

The recovery in Pr/Ph at the core top (Fig. 6), which is observed despite the obvious aggravation of hypoxia until the present, suggests that the distinct minimum in Pr/Ph between 1950 (53.5 cm) and 1990 AD (29 cm) is caused by a post-depositional overprint. Although the mechanism causing a diagenetic decline in Pr/Ph in this depth interval at the present day remains speculative, a likely candidate is the excess Ph production by methanogenic microbes (Brassel et al., 1981; Venkatesan and

Kaplan, 1987; Duan, 2000). Indeed, the current position of the SMTZ at our study site is likely located in the uppermost 10–20 cm below the sediment surface (see Jilbert et al., 2017 and Sawicka and Brüchert, 2017 for comparable systems), below which methane concentrations are expected to increase to the millimolar range. As such, Pr/Ph may not be a direct proxy for bottom water oxygenation at our study site. Instead, it likely records the rate of methanogenesis below the SMTZ, reflecting the amount of labile OM that escapes aerobic degradation. Importantly, the invariably high Pr/Ph prior to the 20[th] century (Fig.

4), could denote that intensive methane formation, due to increased productivity and subsequent shoaling of the redox zonation over the past few decades (Slomp et al., 2013; Egger et al., 2015; Rooze et al., 2016: Jilbert et al., 2017), is unprecedented in our sediment record. However, the low Mo content of < 10 mg kg$^{-1}$ (Fig. 6), suggests that the basin has remained non-euxinic until the present (Scott and Lyons, 2012). This is also evidenced by the slope of ~ 2.7 for the linear regression between Mo and $C_{org}$ (Fig. S1), being slightly shallower than reported for continental margin upwelling systems characterized by

persistently hypoxic, but non-euxinic bottom water conditions (Algeo and Rowe, 2012 and references therein). Furthermore, the MAR of Mo has remained in less than half of the MAR reported for the seasonally mildly euxinic Chesapeake Bay (Helz and Adelson, 2013).



## 7 Implications

Previous studies have suggested that the eutrophication of the Archipelago Sea began in the late 1960s (Bonsdorff et al., 1997a, b; Hänninen et al., 2000; Suomela, 2011). Our data show that environmental conditions in some areas of the Archipelago Sea likely deteriorated several decades prior this, and therefore also prior the establishment of water quality monitoring campaigns

in the 1960s. This highlights the use of sediment-core studies for the long-term reconstruction of environmental conditions in such settings. Our $\delta^{15}N$ record demonstrates increased anthropogenic nutrient input already at the beginning of the 20th century (Fig. 6), although the onset of hypoxia and laminated sediment deposition at this time was likely driven by other factors (Sect. 6.3.2). However, the prominent aggravation of hypoxia in the 1950s was unequivocally anthropogenically-induced. The timing of this shift predates the major establishment period of fish farms in the Archipelago Sea in 1970s (Hänninen et al., 2000),

suggesting that although aquaculture has aggravated hypoxia since the 1970s (Bonsdorff et al., 1997a), other human activities had significantly degraded the coastal water quality up to 20 years previously.

Despite the decreased loading of sewage waters since the 1980s, following the establishment of a wastewater treatment plant for the city of Turku (Suomela, 2011; Fig. 6), the continued leakage of nutrients from agriculture to the Archipelago Sea (Ekholm et al., 2015) together with intensive P regeneration from surface sediments mainly upon the dissolution of Fe-bound

P (Puttonen et al., 2014) have sustained the trend toward increasing eutrophication and shoaling of hypoxia until the present (Figs. 6 and 7). In addition, the recent trajectory towards further aggravation of hypoxia has likely been amplified by the progressively increasing summer temperatures (Fig. 6), which is also supported by the increased importance of climatic effects in the forcing of oxygen depletion in the Swedish coast of the Baltic Sea since the late 1970s (Savage et al., 2010). Hence, while reductions in nutrient loading appear to have improved bottom water oxygenation in the Stockholm Archipelago since

the 1990s (Karlsson et al., 2010), we observe no signs of recovery in the Archipelago Sea so far.

## 8 Conclusions

This study shows that multicentennial-scale climatic oscillations affect near-bottom water oxygenation of a shallow coastal basin in the northern Baltic Sea currently suffering from severe seasonal hypoxia. During warm phases, increased export production of labile, phytoplankton-derived OM combined with effective sediment focusing to the deepest part of the basin

drive deoxygenation of the near-bottom waters in summer. Accordingly, decreased oxygen levels are observed during the MCA and MoWP, but the intensity of the MoWP hypoxia, typified by complete deterioration of the macrobenthic community, is unprecedentedly severe. The progressive deoxygenation during the 1900s was originally triggered by gradual shoaling of the basin due to glacio-isostatic uplift and basin infilling that, together with warming climate, intensified OM delivery primarily through enhanced sediment focusing. Superimposed on these physical factors, exponential increase in anthropogenic nutrient

loading from the catchment stimulated primary productivity and caused a marked aggravation of hypoxia in the 1950s. Our



results demonstrate that the markedly more severe hypoxia during the MoWP in comparison to the MCA is not only attributed to the excess anthropogenic nutrient loading, but also to naturally increased vulnerability of the basin to deoxygenation towards the present. Such natural changes should be considered when elucidating anthropogenic contribution to hypoxia. Furthermore, signs of eutrophication in the area are readily discernible in our sediment record already in the beginning of 1900s, implying

that the water quality diverged from natural conditions decades prior to the establishment of monitoring campaigns. This has important implications for the assessment of reference conditions for water quality in the area. Despite the recent measures taken to reduce anthropogenic nutrient loading to the area, we find no evidence of recovery from hypoxia, suggesting that further measures are needed to alleviate oxygen depletion.

**Data availability**

All data from Figures 1–7 of the manuscript and the Supplementary Figures S1 and S2 will be available in Pangae upon publication of the article.

**Supplement link**

Supplementary Figures S1 and S2 are uploaded together with the manuscript and is available from the Biogeosciences website.

**Author contributions**

SJ devised the study, conducted field and laboratory work, interpreted the data, produced the figures, and drafted the paper. JV devised the study, interpreted the data, and assisted with trace fossil analysis and writing the paper. TJ assisted with laboratory work, interpreted the data, and assisted with writing the paper. JK assisted with the biomarker analyses, interpreted the data, and assisted with writing the paper. OD carried out the ICP-OES and ICP-MS analyses at IOW, interpreted the data, and assisted with writing the paper. HA interpreted the data and assisted with writing paper. JH conducted fieldwork,

interpreted the data and assisted with writing the paper.  LA carried out the IRMS analysis and assisted with writing the paper. MC assisted with the ICP-OES analysis in Helsinki. TS conducted field and laboratory work and assisted with writing the paper.

**Competing interests**

The authors declare that they have no conflict of interest.



**Acknowledgements**

We acknowledge the crew on R/V Aurelia for valuable help in sediment coring. Nadine Hollman, Anne Köhler, Arto Peltola, Jouko Saren, and Hannu Wenho are thanked for their assistance with the laboratory work. Ilppo Vuorinen is thanked for stimulating discussions on the eutrophication of the Archipelago Sea. This research was funded by the Finnish Cultural Foundation project 00150315, Maa- ja Vesitekniikan Tuki project 32719, and by the Baltrap project SAW-2017-IOW-2, funded by the Leibniz Association. SJ received funding from the Doctoral Programme in Biology, Geography and Geology (BGG) at the University of Turku.

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




**Tables**

**Table 1: Retrieved sediment cores.**

| Core ID | Sampling date | Latitude (WGS 84) | Longitude (WGS 84) | Water depth (m) | Gear | Research vessel | Recovery (cm) |
|---|---|---|---|---|---|---|---|
| HAV-KU-5 | 11 June 2015 | 60°14.117N | 22°02.642E | 23.3 | Piston corer | Aurelia | 330 |
| HAV-KU-6 | 11 June 2015 | 60°14.116N | 22°02.642E | 23.3 | Piston corer | Aurelia | 390 |



**Table 2: Classification scheme for assessing bioturbation index. Modified from Behl and Kennett (1996).**

| Bioturbation index | Description of the sedimentary-fabric |
|---|---|
| 1 | Unbioturbated sediment with distinct, continuous lamination |
| 2 | Diffuse, discontinuous or irregular laminations |
| 3 | Slightly bioturbated sediment, with either faint, diffuse laminations/bedding or with few discrete patches of laminations surrounded by homogenized sediment |
| 4 | Completely bioturbated sediment wherein the primary sedimentary-fabric is thoroughly obliterated |





**Table 3: Age constraints given as an input for the age model constructed using Oxcal 4.2 software. Pb pollution features were obtained from Brännvall et al. (1999). The IDs are as in Fig. 2. Measured bulk sediment AMS-$^{14}$C dates are given as $^{14}$C ages without any calibration or reservoir effect correction, although they were not used in the age model due to the large uncertainties in the bulk sediment reservoir effect.**

| Dating method | Event | ID | Core depth (cm) | Age (year AD) | Age uncertainty (±1 σ) |
|---|---|---|---|---|---|
| Varves | | | 3 | 2014 | 5 |
| Varves | | | 13 | 2006 | 5 |
| Varves | | | 23 | 1997 | 5 |
| Cs-137 | Chernobyl | Cs-1 | 33 | 1986 | 2 |
| Varves | | | 43 | 1970 | 5 |
| Varves | | | 53 | 1951 | 5 |
| Varves | | | 63 | 1931 | 5 |
| Varves | | | 73 | 1908 | 5 |
| Varves | | | 76 | 1901 | 5 |
| Pb pollution | Pollution minimum | Pb-1 | 140 | 1600 | 56 |
| Pb pollution | Pollution maximum | Pb-2 | 155 | 1530 | 56 |
| Pb pollution | Pollution minimum | Pb-3 | 179 | 1350 | 50 |
| Pb pollution | Medieval pollution peak | Pb-4 | 219 | 1200 | 71 |
| Pb pollution | Onset of medieval pollution | Pb-5 | 289 | 900 | 50 |
| | | | | | |
| *AMS-$^{14}$C dates* | | | | | |
| Radiocarbon | | Poz-76784 | 72.5 | 725 | 30 |
| Radiocarbon | | Poz-76785 | 147.5 | 670 | 30 |
| Radiocarbon | | Poz-81729 | 174.5 | 660 | 30 |
| Radiocarbon | | Poz-81730 | 212.5 | 670 | 30 |
| Radiocarbon | | Poz-79786 | 229.5 | 270 | 30 |
| Radiocarbon | | Poz-81731 | 260.5 | 270 | 30 |
| Radiocarbon | | Poz-76787 | 304.5 | 60 | 30 |
| Radiocarbon | | Poz-81732 | 350.5 | -15 | 30 |
| Radiocarbon | | Poz-76788 | 389.5 | 15 | 30 |





**Table 4: Spearman rank correlation ($r_s$) matrix for selected variables. The number of stars denotes the degree of significance for each pair of variables: \*\*\* for very high significance ($p < 0.001$), \*\* for high significance ($0.001 < p < 0.01$), \* for significance ($0.01 < p < 0.05$), and no symbol for weak or no significance ($p < 0.05$). Correlation coefficients with absolute values exceeding 0.70 are bolded.**

| | $C_{org}$ | C/N | $C_{org}$ MAR | BIT | Mo | Mo MAR | Pr/Ph | Ti/K | $\delta^{13}C_{org}$ | $\delta^{15}N$ |
|---|---|---|---|---|---|---|---|---|---|---|
| $C_{org}$ | 1 | \*\*\* | \*\*\* | \*\*\* | \* | \*\*\* | - | \*\*\* | \*\* | - |
| C/N | **-0.73** | 1 | \*\*\* | \*\*\* | \*\* | - | \*\*\* | \*\*\* | \*\* | - |
| $C_{org}$ MAR | **0.70** | -0.34 | 1 | \*\* | \*\*\* | \*\*\* | \* | \*\*\* | - | \*\* |
| BIT | **-0.74** | **0.76** | -0.38 | 1 | - | - | \*\*\* | \*\*\* | \*\*\* | - |
| Mo | 0.18 | 0.23 | 0.54 | 0.07 | 1 | \*\*\* | \*\*\* | \* | - | - |
| Mo MAR | 0.37 | 0.04 | **0.80** | -0.03 | **0.91** | 1 | \*\*\* | \*\*\* | - | \*\* |
| Pr/Ph | 0.20 | -0.52 | -0.30 | -0.41 | -0.48 | -0.47 | 1 | - | \* | \* |
| Ti/K | **0.83** | -0.53 | **0.80** | -0.68 | 0.40 | 0.58 | 0.05 | 1 | \*\*\* | - |
| $\delta^{13}C_{org}$ | 0.51 | -0.52 | 0.09 | **-0.76** | -0.12 | -0.14 | 0.43 | 0.63 | 1 | \*\* |
| $\delta^{15}N$ | 0.12 | 0.12 | 0.46 | 0.22 | 0.34 | 0.47 | -0.44 | 0.04 | -0.54 | 1 |

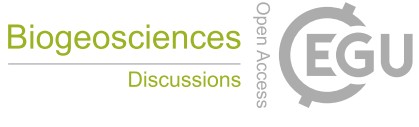

**Figures**

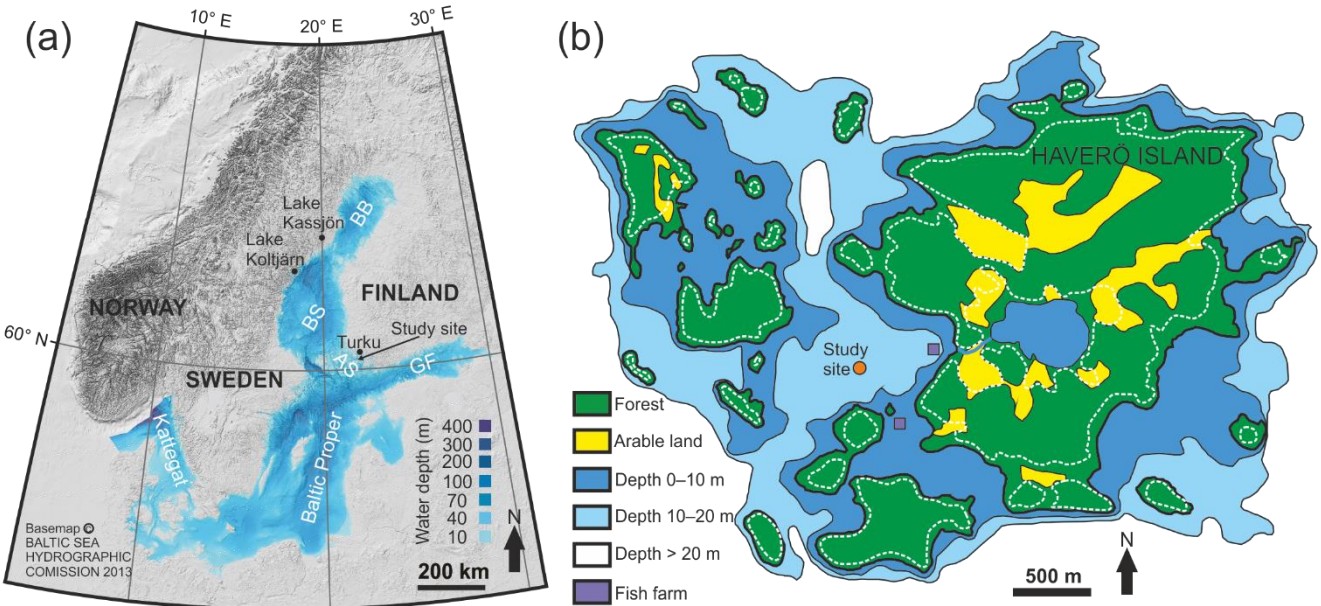

**Figure 1: Maps of the study area. (a) Bathymetric map of the Baltic Sea. The abbreviations for the indicated sub-basins are as**
5 **follows: BB = Bothnian Bay, BS = Bothnian Sea, AS = Archipelago Sea, GF = Gulf of Finland. Locations of the varved lakes used**
**for the correlation of the pollution Pb profiles are also shown. B) Inset map of the study location. White dashed line indicates the 5**
**m contour line that roughly corresponds to the paleoshoreline at ~ 760 AD.**





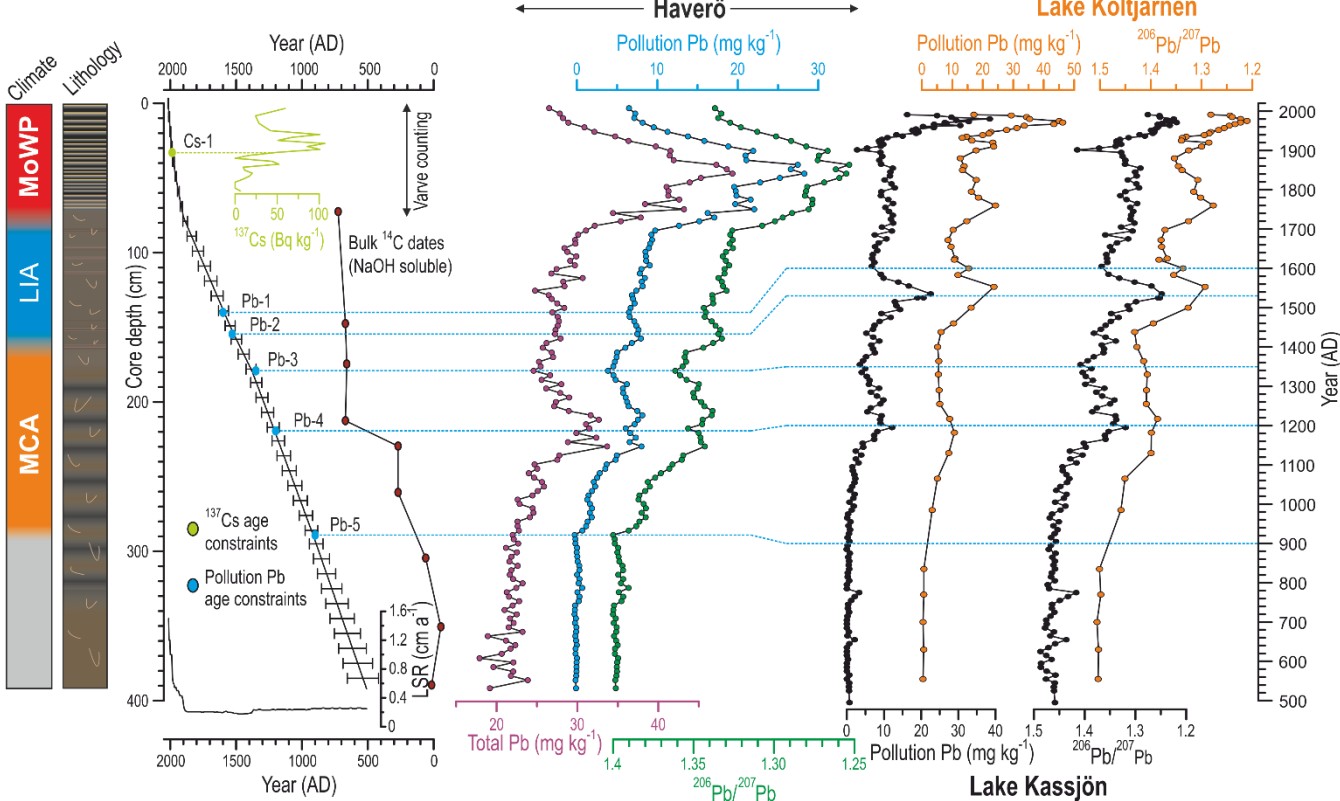

**Figure 2: Maps of the study area. (a) Bathymetric map of the Baltic Sea. The abbreviations for the indicated sub-basins are as follows: BB = Bothnian Bay, BS = Bothnian Sea, AS = Archipelago Sea, GF = Gulf of Finland. Locations of the varved lakes used for the correlation of the pollution Pb profiles are also shown. B) Inset map of the study location. White dashed line indicates the 5**
5 **m contour line that roughly corresponds to the paleoshoreline at ~ 760 AD.**



**Figure 3: X-radiographs of different lithological units together with their interpretations. (a)** Intensely *Planolites*-mottled sedimentary-fabric characterized by a low number of distinct traces. **(b)** Thinly bedded sedimentary-fabric overprinted by abundant *Planolites* trace fossils. The lowermost bed is disturbed by bivalve biodeformation (bbd). **(c)** Intensely burrow-mottled sedimentary-fabric with abundant *Planolites* (Pl) and *Arenicolites* (Ar) trace fossils. **(d)** Sharply laminated sedimentary-fabric punctuated with gentle blurring of the lamination. The black holes in the X-radiographs are artefacts produced by holes in the sample boxes.



**Figure 4:** Profiles of median grain size and Ti/K denoting fluctuations in local hydrodynamic conditions plotted together with the proxies for hypoxia intensity.





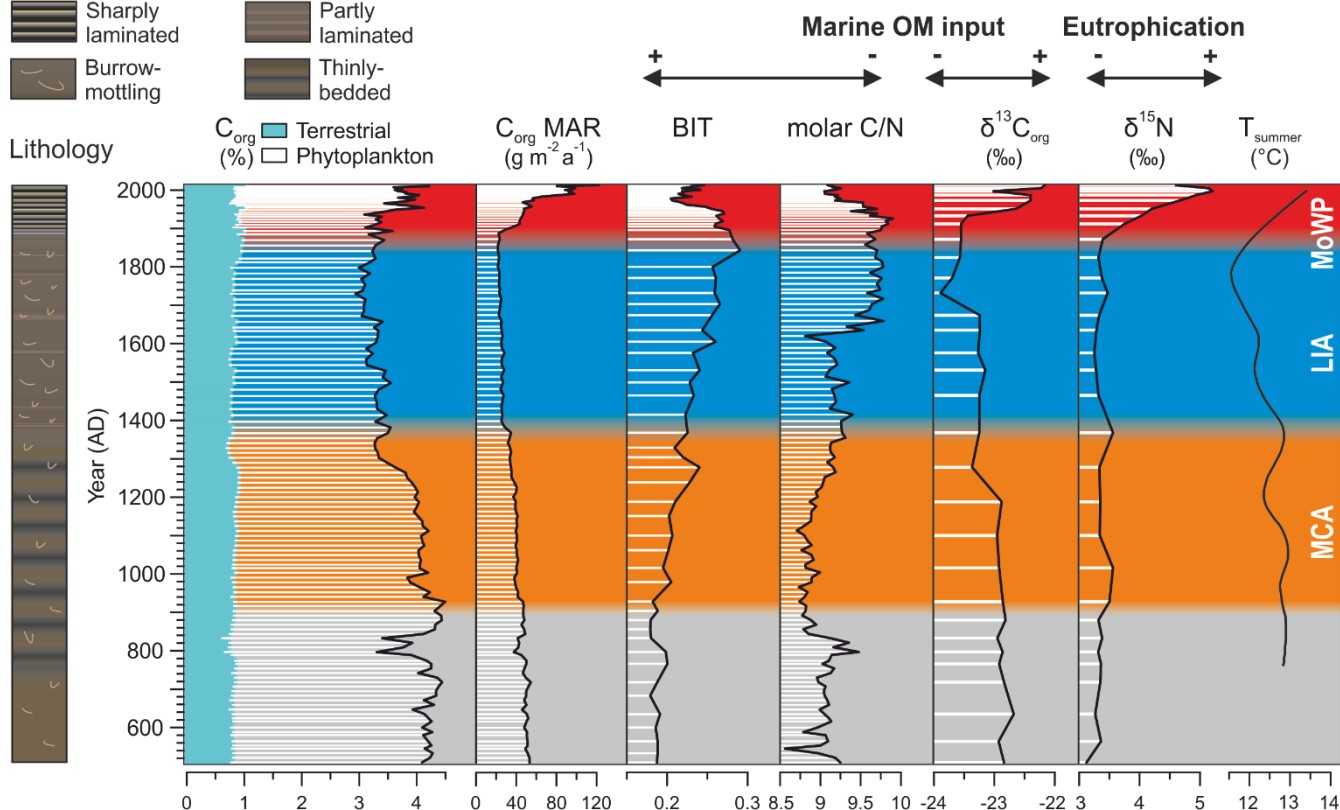

**Figure 5: Geochemical profiles reflecting the delivery and preservation of organic material in the basin. Fractions of C$_{org}$ were calculated from molar N/C ratios, applying end-member values of 0.04 and 0.13 for terrestrial and phytoplankton-derived C$_{org}$, respectively. A loess-smoothed dendroclimatic summer temperature reconstruction for the south-eastern Finland (data from Helama et al., 2014) is also shown. Note the marked decline in the input of phytoplankton-derived C$_{org}$ during the LIA.**









**Figure 6: Comparison of possible climatic and anthropogenic drivers of eutrophication in the study area versus geochemical profiles reflecting the sources and delivery organic matter to the sediment ($C_{org}$, $\delta^{13}C_{org}$, and $\delta^{15}N$) and redox conditions at the sediment-water interface (content and MAR of Mo, Pr/Ph ratio) over the past ~ 200 years. Modeled development of hypoxic area in the Baltic Sea (redrawn from Carstensen et al., 2014b), excluding Kattegat, Danish Straits, and coastal zone, is shown for reference. Data regarding agricultural practices in Finland (annual sales of N and P fertilizers and area of cultivated land) were obtained from the Natural Resources Institute of Finland. Maximum ice extent data for the Baltic Sea (MIB) was provided by the Finnish Meteorological Institute (30-year moving average with error range indicated for the early estimates), and summer temperature record for the city of Turku (shown as 11-year running mean) was retrieved from Tuomenvirta et al. (2015). The population growth curve for the city Turku was compiled from Lahtinen (2014) and data provided by the Population Register Centre of Finland.**



**Figure 7: Contour plots for water column dissolved oxygen concentration through time at four intensive monitoring stations (data from HERTTA database) located around the study site (HAV-KU-6). The interpolations were produced with the Ocean Data View software (Schlitzer, 2017). White dots represent the original measurement data.**