# Peer review of "● MoWP ● LIA ● MCA ○ pre-MCA"

_Biogeosciences, 2018_

## Referee Comment (RC1) · Anonymous Referee #1 · 21 Feb 2018

The authors presented a 1500-year multiproxy sedimentary record form the Archipelago Sea in the Baltic Proper. The records show a progressive eutrophication in the region and a pronounced aggravation of bottom water hypoxia in the 1950s, which is unprecedented. This is interesting and merits publication. However, the authors fail to convince me that between 900 and 1900 the multicentennial variability of the bottom water oxygen concentration was locally forced. Furthermore, it remained puzzling how the combined effects of gradual shoaling of the basin and warming climate amplified sediment focusing and increased the vulnerability to hypoxia. This however appears to be crucial to authors since they conclude that: 'such natural changes should be considered when elucidating anthropogenic contribution to hypoxia' I would suggest to

include C/N ratios, d13C and d15N values of soil organic matter into discussion and most importantly into the mixing analyses used to quantify inputs of terrestrial organic matter.

Please find further comments below:

Authors: (Fig 5) Indeed, despite the negative long-term trend in autochthonous OM concentration from the pre-MCA towards the present, the MCA and MoWP are typified by relatively high input in comparison to the LIA, implying enhanced productivity under warm climatic phases.

This is difficult to see in the figure 5. If at all the declining OM concentration hold on until about 1700 and not towards the present. Assuming that OM% indicates primary production why OM% during the MoWP does not exceed those during the MCA despite heavy eutrophication and global warming after 1900.

Authors: By contrast, a distinct decrease in Corg and $\delta$13 Corg , paralleled by an increase in C/N and BIT index, coincides with the MCA– LIA transition (Fig. 5), suggesting that this marked decline in local primary productivity was most likely forced by the climatic cooling (Kabel et al., 2012).

The use of OM% as productivity indicator and ignoring Corg MAR is problematic and calls for further explanation justifying this interpretation.

Page 20 Authors: However, this period is characterized by a shift to suppressed primary productivity at our study site, implying no influence of enhanced external nutrient inputs.

Decreasing OM% and increasing BIT implies to me that an enhanced input from land could have lower OM% because of dilution. Since at this time grain sizes decrease this might be a response to enhanced aeolian dust inputs due to the expanding agriculture at around 1400. Which role could the changing sediment structure play with respect to the preservation of OM in the sediment?

Authors: The lack of anthropogenic forcing of OM input during the MCA is also evidenced by the constant sediment $\delta15$ N signature over this period at Haverö (Fig. 5; see also Cole et al., 2004).

Impacts on d15N I would expect only in response to an intensive use of fertilizer which occurred much later as indicate by the provided data. I would expect an enhanced soil erosion, that is why I suggested to integrate C/N ratios and $\delta13$ Corg of soil organic matter.

Page 20 Authors: Yet, we observe negligible changes in the source of OM around 1900 AD, as implied by the relatively constant C/N and $\delta13$ C values (Fig. 6). This suggests that the strong increase in the Corg MAR at this time was mainly caused by intensified sediment focusing, as supported by the contemporaneous increases in Ti/K and sediment MAR (Figs. 4 and 6). We propose that the general trend towards higher source-to-sink ratio in the basin, combined with the climate-driven intensification of wind-induced sediment reworking (Fig. 6) to increase the Corg MAR to the sediments

I would argue that a stronger physical forcing in addition to waste water discharges (enriched in 15N) increased the marine productivity which decreased BIT, C/N ratios and 13C values as seen in your data.

Authors: By contrast, the shift to unprecedentedly heavy $\delta13$ C signature, sympathetic with a decrease in C/N and BIT index around 1950 AD (Fig. 6), points to direct enhanced export production of phytoplankton-derived OM at this time (Meyers, 1994; 1997; Hopmans et al., 2004).

I agree but this occurred already before but on a lower scale.

Page 21 Authors: We attribute the multicentennial-scale fluctuations in bottom water oxygenation associated with the MCA and LIA to climatic variability that modulated both hydrographic conditions and accumulation of OM at the sea floor.

It is not clear on which data this statement based.

Page 23 Authors: Considering the negligible variation in the proxies for the source of OM prior to 1930 AD (Fig. 6), the onset of seasonal hypoxia and the resulting preservation of continuous lamination since the beginning of 20th century was apparently not forced by changes in primary productivity. Instead, we postulate that this deoxygenation was forced 5 by the following complex interplay of warming climate and millennial-scale changes in the basin configuration:

To me your data are showing that eutrophication increases primary production and hypoxia.

Page 23 Authors: Accordingly, although we can not completely exclude the possible contribution of anthropogenic forcing, the onset of recurring seasonal hypoxia at around 1900 AD can be largely attributed to the naturally increased vulnerability to deoxygenation that together with global warming irreversibly tipped the ecosystem over a threshold, inducing a regime shift commonly associated with coastal oxygen deficiency (e.g. Conley et al., 2009b).

I would say the opposite: anthropogenic forcing seems to be the decisive factor but you cannot rule out others processes.

Page 24 Authors: We postulate that, in addition to effective sediment focusing, the increased sediment MAR was likely fueled by ballasting effects, whereby eutrophication-induced increase in the OM production in the euphotic zone drives the sedimentation of fine-grained lithogenic material through aggregation (Passow, 2004; Passow and De La Rocha, 2006; De La Rocha et al., 2008), as suggested by the close covariation between sediment and Corg MARs (Fig. 6).

This part sound also odd: What fuels the ballast effect?

Page 25 Authors: Our data show that environmental conditions in some areas of the Archipelago Sea likely deteriorated several decades prior this, and therefore also prior the establishment of water quality monitoring campaigns in the 1960s.

Not at some areas only at your study sites.

Our $\delta$15N record demonstrates increased anthropogenic nutrient input already at the beginning of the 20th century (Fig. 6), although the onset of hypoxia and laminated sediment deposition at this time was likely driven by other factors (Sect.6.3.2).

If you would replace 'likely' with 'additionally' I would almost agree.

Page 25 Conclusion Authors: This study shows that multicentennial-scale climatic oscillations affect near-bottom water oxygenation of a shallow coastal basin in the northern Baltic Sea currently suffering from severe seasonal hypoxia. During warm phases, increased export production of labile, phytoplankton-derived OM combined with effective sediment focusing to the deepest part of the basin drive deoxygenation of the near-bottom waters in summer.

Based which data you define 'labile' To my understanding 'sediment focusing' should favor the accumulation of more refractory OM and considering the declining CorgMAR until 1900 I doubt that changes in bottom water concentration are locally forced.

Authors: The progressive deoxygenation during the 1900s was originally triggered by gradual shoaling of the basin due to glacio-isostatic uplift and basin infilling that, together with warming climate, intensified OM delivery primarily through enhanced sediment focusing.

Please clarify: What fills the basin, form where the filling comes what do you expect climate to do!

---

## Referee Comment (RC2) · Anonymous Referee #2 · 15 Mar 2018

General Comments: The paper by Joniken et al. is a multiproxy study of the development of coastal hypoxia in the Archipelago Sea, Northern Baltic Sea during the last millenium. It takes into consideration the natural changes in the basin geomorphology and sediment transport, driven by climate variability, with the anthropogenic inputs, as causes of the emergence of near-bottom hypoxia in the area. One of the main findings is that coastal hypoxia started to occur in the early 1900's, and not in the mid-twentieth century, as other studies have suggested. The authors state that natural variability and trends were the main drivers of the emergence of hypoxia in the 1900's, but that eutrophication was the key factor from the 1950's onwards. The paper is well-written, including many data in tables and figures. Perhaps alternative interpretations on the

studied record are missing, but, overall, he manuscript deserves to be accepted for publication. Minor revisions are suggested as follows.

Specific comments: 1) The'material and methods' section is too long and might be simplified. For example, subsection 4.4, could be summarized and the full text be moved to supplementary material.

2) p.22 The authors propose that Mo accumulation rate is a proxy for bottom water hypoxia in the MoWP, suggesting that this takes place mostly during summer. Nevertheless, Mo geochemistry depends on the suplhide concentration in the pore waters, which in turn also depends on the sedimentation rate of labile organic matter. As the authors recognize, the phytoplankton productive season extends during spring/summer. Thus Mo accumulation rate does not actually depend on bottom water hypoxia, rather the latter be a consequence of organic matter respiration in the surface sediments. This should be taken into account in the discussion. On other hand, in order to preclude any bias due to the changes in the sedimentation rate for the authigenic accumulation of Mo, it would be better to normalize the Mo content to Aluminum, and estimate the Mo enrichment factor (Scholz et al., 2013).

3) p.23 on causes of early 1900s deoxygenation: the authors include stratification and global warming as one of the factors precluding the bottom water ventilation and the deoxygenation trend. However, SST paleorecords (Figure 5), shows that the warming is significant (above that attained in the MCA) just around 1950, while laminations, increased d15N and increased organic carbon and Mo accumulation rates are recorded well before.

4) The attribution to 'warmer climate' for the early deoxygenation is also mentioned in the conclusions. Authors might reconsider this interpretation or provide more support to this conclusion.

5) p.20. On the role of the sewer network for the city which likely enhanced sewage loading to the Archipelago Sea, at the beginning of the twentieth century, the authors

indicate that it had a secondary role, because no significant changes in the organic matter source were detected. However this argument is not convincing, since the increases of MAR, OC MAR and d15N occurred just after the sewer construction, and a higher influence of marine organic matter was recorded later, driving the exacerbation of hypoxia. It would be possible that anthropogenic organic enrichment partly sustained the early increase of MAR and also contributed the increase of primary productivity in the area.

5) p.25 ('Implications'). The authors indicate that the $\delta$15N record demonstrates increased anthropogenic nutrient input already at the beginning of the 20th century. It would be worth to analyze if the record is alsorelated to an increased denitrification in the bottom waters and surface sediments, which could result of an increasing anthropogenic/natural labile organic matter flux.

6) Figure 7. Though all the figures coincide in showing a negative trend of near-bottom water oxygenation; it would have been better to provide a climatology (or monthly/seasonal averages) of the water column dissolved oxygen concentration. The resolution of the time-series does not allow to observe the seasonal hypoxia.

Technical comment: The legend of Figure 2 (geochronologies) is missing (the text of the legend 1 was duplicated here).
* * *
**BGD**

---

## Author Comment (AC1) · 5 Apr 2018

Below we will address the comments from both referees with the following sequence:

**[Referee comments in bold]**

*[Responses in italics]*

[Changes in the manuscript in regular text]

**1. Responses to online discussion Referee #1**

**The authors presented a 1500-year multiproxy sedimentary record form the Archipelago Sea in the Baltic**
**Proper. The records show a progressive eutrophication in the region and a pronounced aggravation of**
**bottom water hypoxia in the 1950s, which is unprecedented. This is interesting and merits publication.**
**However, the authors fail to convince me that between 900 and 1900 the multicentennial variability of the**
**bottom water oxygen concentration was locally forced. Furthermore, it remained puzzling how the combined**
**effects of gradual shoaling of the basin and warming climate amplified sediment focusing and increased the**
**vulnerability to hypoxia. This however appears to be crucial to authors since they conclude that: 'such**
**natural changes should be considered when elucidating anthropogenic contribution to hypoxia' I would**
**suggest to include C/N ratios, d13C and d15N values of soil organic matter into discussion and most**
**importantly into the mixing analyses used to quantify inputs of terrestrial organic matter.**

*We thank the referee for thorough consideration of our manuscript. Below we address each of the specific*
*comments.*

**Page 19**

**Authors: (Fig 5) Indeed, despite the negative long-term trend in autochthonous OM concentration from the**
**pre-MCA towards the present, the MCA and MoWP are typified by relatively high input in comparison to**
**the LIA, implying enhanced productivity under warm climatic phases.**

**This is difficult to see in the figure 5. If at all the declining OM concentration hold on until about 1700 and**
**not towards the present. Assuming that OM% indicates primary production why OM% during the MoWP**
**does not exceed those during the MCA despite heavy eutrophication and global warming after 1900.**

*Indeed, the existence of a long-term trend in $C_{org}$ is somewhat debatable, while BIT, C/N and $\delta^{13}C_{org}$ are showing*
*this trend more clearly. The $C_{org}$ content during the MoWP remains at the MCA level most likely due to the dilution*
*effect, resulting from the combined effects of intensive sediment focusing and ballasting during the MoWP (Sections*
*6.2.2 and 6.3.2). This is in accordance with the unprecedentedly high sediment and $C_{org}$ MAR during the MoWP*
*(Fig. 5).*

We will exclude the statement about the long-term negative trend in $C_{org}$ and emphasize the dilution of $C_{org}$ content
during the MoWP.

**Authors: By contrast, a distinct decrease in Corg and δ13 Corg , paralleled by an increase in C/N and BIT**
**index, coincides with the MCA– LIA transition (Fig. 5), suggesting that this marked decline in local primary**
**productivity was most likely forced by the climatic cooling (Kabel et al., 2012).**

**The use of OM% as productivity indicator and ignoring Corg MAR is problematic and calls for further**
**explanation justifying this interpretation.**

*We agree that the use $C_{org}$ content as a proxy for productivity is problematic due to the potential dilution and*
*preservation issues. However, in this case, all other proxies for the delivery of OM (BIT, C/N and $\delta^{13}C_{org}$) support*
*our interpretation of decreased delivery of phytoplankton-derived OM at the MCA-LIA transition, suggesting that*
*the contemporaneous decline in $C_{org}$ was likely attributed to decreased productivity in the photic zone. We also*
*note that there is a subtle decrease in $C_{org}$ MAR (Fig. 5) and LSR (Fig. 1) at the MCA-LIA transition (Fig. 5), but*
*the unprecedentedly high $C_{org}$ MAR during the MoWP is masking this. However, we acknowledge that in this lower*
*part of the core, the scarcity of age constraints hampers recognition of potential rapid changes in the sedimentation*
*rate.*

We will stress even more explicitly that all of the proxies for the OM delivery coincide, and that the $C_{org}$ profile
alone can not be used to assess changes in primary productivity. In addition, we will clarify the potential issues
related to linking $C_{org}$ content (dilution effects) and MAR (sediment focusing) to primary productivity. We will
also add a notion that our MAR profiles are not able to pick up potential decadal (or shorter) scale fluctuations
prior to the 1900s due to the lack of age constraints.

**Page 20 Authors: However, this period is characterized by a shift to suppressed primary productivity at our**
**study site, implying no influence of enhanced external nutrient inputs.**

**Decreasing OM% and increasing BIT implies to me that an enhanced input from land could have lower**
**OM% because of dilution. Since at this time grain sizes decrease this might be a response to enhanced aeolian**
**dust inputs due to the expanding agriculture at around 1400. Which role could the changing sediment**
**structure play with respect to the preservation of OM in the sediment?**

*Aeolian dust input is not a significant vector of sedimentation in this setting. Our previous study on seasonal*
*sedimentation dynamics in this basin (Jokinen et al., Mar. Geol. 366, 2015) clearly shows that the sedimentation*
*of lithogenic material is largely controlled by the intensive wave-induced reworking and subsequent lateral*
*transport of previously deposited late- and post glacial clays and brackish-water muds linked to the glacio-isostatic*
*uplift. There are no signs of vernal riverine sediment plumes (spring thaw) affecting the sedimentation of lithogenic*
*material (Jokinen et al., 2015). This is already stated in the description of the study location in Section 2.*
*Furthermore, as indicated by our Ti/K and grain size profiles, the sediment reworking which has the potential to*

*dilute the $C_{org}$, content, was at maximum during the warm phases of the MCA and MoWP (Fig. 4). Moreover, we*
*stand by the assertion that our binary mixing model for C/N robustly tracks past variations in the delivery of*
*terrestrial vs. phytoplankton-derived OM in this setting (see our later response), and suggests negligible changes*
*in the input of terrestrial OM throughout the record (Fig. 5). By contrast, the mixing model shows that the changes*
*in $C_{org}$ content are controlled by changes in the delivery of phytoplankton-derived OM, which was suppressed*
*during the LIA. This interpretation is independently supported by the the BIT and $\delta^{13}C_{org}$ profiles. As for the relation*
*between the change in sedimentary-fabric and OM preservation we again refer to the consistency between BIT,*
*C/N, $\delta^{13}C_{org}$ and $C_{org}$ profiles, all of which are pointing to a decrease in the delivery of phytoplankton-derived OM*
*at the MCA-LIA transition. Yet, we acknowledge that the $C_{org}$ accumulation in sediment is an integral of OM*
*delivery and preservation. However, as these two mechanisms often work in concert, we infer that although the*
*increased $C_{org}$ content during the warm phases might be partly attributed enhanced preservation due to decreased*
*oxygen availability, it is likely that the decrease in near-bottom water oxygenation was primarily forced by*
*intensified OM delivery (e.g. Pedersen and Calvert, Bull. Am. Assoc. Petrol. Geol. 74, 1990). Finally, we note that*
*a recent study from the eastern coast of Sweden, where intensification of land use likely occurred earlier than in*
*our study region, suggests that the onset of cultural eutrophication became notable in the coastal sediment record*
*not earlier than in the 1800s (Ning et al., Anthropocene 21, 2018).*

For the reasons described above, we retain the interpretation that the delivery of OM in the basin is largely
controlled by the local primary productivity, which declined during the LIA as a response to climatic cooling. We
will clarify that sediment OM accumulation is an integral of OM delivery and preservation, whereby increased
delivery is likely to enhance preservation through decreased bottom-water oxygenation.

**Authors: The lack of anthropogenic forcing of OM input during the MCA is also evidenced by the constant**
**sediment δ15 N signature over this period at Haverö (Fig. 5; see also Cole et al., 2004).**

**Impacts on d15N I would expect only in response to an intensive use of fertilizer which occurred much later**
**as indicate by the provided data. I would expect an enhanced soil erosion, that is why I suggested to integrate**
**C/N ratios and δ13 Corg of soil organic matter.**

*We thank the referee for pointing out that the constant $\delta^{15}N$ over the MCA does not provide sufficient evidence for*
*the argument. As for the inclusion of soil organic matter C/N ratios and $\delta^{13}C_{org}$ values into the mixing model, we*
*thank the referee for the suggestion, and acknowledge that we did not point out our assumptions related to the*
*binary mixing model explicitly enough. Following Goñi et al. (Est. Coast. Shelf S. 57, 2003) and Jilbert et al.*
*(Biogeosciences 15, 2018), our terrestrial end-member integrates both soil and plant-derived OM, because*
*essentially all of the OM transported by rivers to the coastal zone passes through the soil reservoir, and ranges in*
*composition from relatively fresh vascular plant detritus to more degraded soil OM. Majority of this riverine OM*
*input to the estuaries along the coastal areas of Finland occurs in dissolved form (see Jilbert et al., 2018 and*
*references therein).  Importantly, the two end-member mixing model applied in our study has been shown to be a*
*functional tool in estimating the relative contribution of terrestrial versus phytoplankton-derived OM in an*

*analogous setting along an estuary in the northern Baltic Sea coast (Jilbert et al., Biogeosciences 15, 2018). This*
*is supported by a crossplot of C/N against BIT (Fig. R1), where the linear coupling between the parameters with*
*no marked deviations from the regression line suggests that our two end-member mixing model is sufficient to*
*describe the system, and thus inclusion of a separate third end-member is not necessary. Furthermore, the close*
*correlation between C/N and BIT (Fig. R1) suggests that either the C/N ratios of soil and plant OM are rather*
*similar or that the relative contribution of these two compartments has remained more or less constant. We also*
*note that there is considerable uncertainty related to the selection of a valid end-member C/N ratio for soil OM*
*(See e.g. Weijers et al., GCA 73, 2009), meaning that the results of a three end-member mixing model, wherein the*
*soil OM is incorporated as a separate end-member, would be highly sensitive to this rather arbitrary selection.*
*Finally, we again refer to the marked consistency between all the proxies for the delivery of OM (BIT, C/N and*
*$\delta^{13}C_{org}$) in our sediment record. Based on these considerations, we infer that it is more objective to follow the binary*
*mixing model by Jilbert el al. (2018) that has been shown to reliably trace the contribution of terrestrial OM in*
*relation to phytoplankton-derived OM in a similar setting.*

We will exclude the argument regarding constant $\delta^{15}$N signature over the MCA from the revised version of the
manuscript. However, we retain the two end-member mixing model for C/N, but clarify more explicitly that the
terrestrial OM compartment integrates both soil and plant OM.

[Figure]

Fig. R1. A crossplot between C/N and BIT index.

**Page 20 Authors: Yet, we observe negligible changes in the source of OM around 1900 AD, as implied by the relatively constant C/N and δ13 C values (Fig. 6). This suggests that the strong increase in the Corg MAR at this time was mainly caused by intensified sediment focusing, as supported by the contemporaneous increases in Ti/K and sediment MAR (Figs. 4 and 6). We propose that the general trend towards higher source-to-sink ratio in the basin, combined with the climate-driven intensification of wind-induced sediment reworking (Fig. 6) to increase the Corg MAR to the sediments.**

**I would argue that a stronger physical forcing in addition to waste water discharges (enriched in 15N) increased the marine productivity which decreased BIT, C/N ratios and 13C values as seen in your data.**

*Both reviewers suggest that the role of anthropogenic forcing already in the beginning of $20^{th}$ century should be stressed. However, in our opinion it is an oversimplification to state that the increase in OM input around 1900 AD was only related to human-induced eutrophication. Accordingly, the constant C/N, BIT and $\delta^{13}C_{org}$ over the transition to enhanced sediment and OM accumulation (and onset of recurring seasonal hypoxia) at the onset of $20^{th}$ century should not be ignored. However, after careful reconsideration, we admit that the anthropogenic forcing in the early 1900s due to the population growth and sewage network construction merits more attention as a driver of intensified OM deposition (and subsequent aggravation of hypoxia) as suggested by the similarity between $\delta^{15}$N, $C_{org}$ MAR and Mo MAR profiles and the population growth in Turku (Fig. 6). Still, we stand by the assertion that the physical factors that enhanced sediment focusing, stratification, and natural vulnerability to hypoxia have to be acknowledged in the manuscript because the depositional conditions unarguably have changed through time due to climatic oscillations and changes in the physical configuration of the basin (e.g. Section 6.3.2). Combined, we infer that the regime shift around 1900 AD most likely resulted from the combined effects of increased physical and anthropogenic forcing that together irreversibly tipped the system over a threshold and initiated the feedback mechanisms associated with hypoxia.*

We will rephrase the revised version of the manuscript to emphasize that the increase in OM accumulation and onset of recurring seasonal hypoxia since around 1900 AD was triggered by a combination of physical and anthropogenic forcing factors, instead of attributing this regime shift merely to physical factors.

**Authors: By contrast, the shift to unprecedentedly heavy δ13 C signature, sympathetic with a decrease in C/N and BIT index around 1950 AD (Fig. 6), points to direct enhanced export production of phytoplankton-derived OM at this time (Meyers, 1994; 1997; Hopmans et al., 2004).**

**I agree but this occurred already before but on a lower scale.**

*We partly agree. See our previous response.*

We will rephrase the sentence accordingly to state that the early increase in OM delivery (1900 AD) was attributed to a combination of physical and anthropogenic forcing factors, whereas the further intensification of OM

accumulation (and aggravation of hypoxia) around 1950 AD was more closely linked to the drastically increased
human-induced nutrient loading.

**Page 21 Authors: We attribute the multicentennial-scale fluctuations in bottom water oxygenation**
**associated with the MCA and LIA to climatic variability that modulated both hydrographic conditions and**
**accumulation of OM at the sea floor.**

**It is not clear on which data this statement based.**

*In our opinion, the basis for the change in hydrographic conditions and accumulation of OM is clearly stated in*
*the following sentence: "This is supported by the Ti/K and grain size profiles and the organic matter proxies*
*(Sections. 6.1 and 6.2) that indicate amplified lateral sediment transport (focusing) and primary productivity*
*during the MCA in comparison to the LIA (Figs. 4 and 5). In case the reviewer means that the basis for the*
*multicentennial-scale fluctuations in bottom water oxygenation is not clearly stated, we would like to note that this*
*is rather explicitly pointed out earlier in Section 6.3.1.*

We find no clear reasons to rephrase this sentence.

**Page 23 Authors: Considering the negligible variation in the proxies for the source of OM prior to 1930 AD**
**(Fig. 6), the onset of seasonal hypoxia and the resulting preservation of continuous lamination since the**
**beginning of 20th century was apparently not forced by changes in primary productivity. Instead, we**
**postulate that this deoxygenation was forced by the following complex interplay of warming climate and**
**millennial-scale changes in the basin configuration:**

**To me your data are showing that eutrophication increases primary production and hypoxia.**

*Partly agreed. See our previous response. We infer that this regime shift resulted from a combination of physical*
*and anthropogenic forcing that together enhanced OM accumulation and the vulnerability of the basin to hypoxia.*

We will ascribe the early deoxygenation in the early 1900s more closely to human-induced eutrophication, which
might have stimulated primary productivity and, together with physical forcing, irreversibly tipped the system to
seasonal hypoxia.

**Page 23 Authors: Accordingly, although we can not completely exclude the possible contribution of**
**anthropogenic forcing, the onset of recurring seasonal hypoxia at around 1900 AD can be largely attributed**
**to the naturally increased vulnerability to deoxygenation that together with global warming irreversibly**
**tipped the ecosystem over a threshold, inducing a regime shift commonly associated with coastal oxygen**
**deficiency (e.g. Conley et al., 2009b).**

**I would say the opposite: anthropogenic forcing seems to be the decisive factor but you cannot rule out others**
**processes.**

*Partly agreed. See our previous responses.*

We will adjust the interpretation according to the points made above.

**Page 24 Authors: We postulate that, in addition to effective sediment focusing, the increased sediment MAR**
**was likely fueled by ballasting effects, whereby eutrophication-induced increase in the OM production in**
**the euphotic zone drives the sedimentation of fine-grained lithogenic material through aggregation (Passow,**
**2004; Passow and De La Rocha, 2006; De La Rocha et al., 2008), as suggested by the close covariation**
**between sediment and Corg MARs (Fig. 6).**

**This part sound also odd: What fuels the ballast effect?**

*In our opinion, this process is rather explicitly explained in the quoted and in the following sentence: "Indeed, it*
*has been shown that rapid sedimentation events during vernal phytoplankton blooms in the study area are caused*
*by the formation of organomineralic aggregates adhered together by sticky transparent exopolymers (TEP)*
*excreted by phytoplankton (Jokinen et al., 2015).*

We will further clarify the ballasting mechanism in the revised version of the manuscript.

**Page 25 Authors: Our data show that environmental conditions in some areas of the Archipelago Sea likely**
**deteriorated several decades prior this, and therefore also prior the establishment of water quality**
**monitoring campaigns in the 1960s.**

**Not at some areas only at your study sites.**

*We thank the referee for pointing out this vague statement.*

We will adjust the statement as suggested.

**Our δ15N record demonstrates increased anthropogenic nutrient input already at the beginning of the 20th**
**century (Fig. 6), although the onset of hypoxia and laminated sediment deposition at this time was likely**
**driven by other factors (Sect. 6.3.2).**

**If you would replace 'likely' with 'additionally' I would almost agree.**

*Partly agreed. See our previous responses.*

Again, we will rephrase the statement to emphasize the combined effects of physical and anthropogenic forcing.

**Page 25 Conclusion Authors: This study shows that multicentennial-scale climatic oscillations affect near-bottom water oxygenation of a shallow coastal basin in the northern Baltic Sea currently suffering from severe seasonal hypoxia. During warm phases, increased export production of labile, phytoplankton-derived OM combined with effective sediment focusing to the deepest part of the basin drive deoxygenation of the near-bottom waters in summer.**

**Based which data you define 'labile' To my understanding 'sediment focusing' should favor the accumulation of more refractory OM and considering the declining CorgMAR until 1900 I doubt that changes in bottom water concentration are locally forced.**

*We infer it is likely that increased sediment focusing will enhance the accumulation of both labile and refractory OM. Considering the combined effects of enhanced sediment focusing and intensified primary productivity during warm climatic phases, it is reasonable to expect that not only refractory but also the fresh, phytoplankton-derived OM will be more effectively buried below the zone of active bioturbation. As for the $C_{org}$ MAR, see our previous response.*

We will rephrase the revised manuscript to emphasize that the intensified sediment focusing during the MCA and MoWP enhanced the burial of both labile and refractory OM.

**Authors: The progressive deoxygenation during the 1900s was originally triggered by gradual shoaling of the basin due to glacio-isostatic uplift and basin infilling that, together with warming climate, intensified OM delivery primarily through enhanced sediment focusing.**

**Please clarify: What fills the basin, form where the filling comes what do you expect climate to do!.**

*In our opinion, we have explicitly described how the depositional system works in response to climatic oscillations. First, in Section. 6.1 we describe the combined effects of glacio-isostatic uplift and sediment infilling, and superimposed climatic oscillations on the depositional setting. Specifically, we postulate that the sediment focusing is more effective during the warm phases when wave-induced sediment reworking is more effective. Then, we show that under warm climatic phases the delivery of phytoplankton-derived OM (which largely controls the OM accumulation) increases, which together with effective sediment focusing leads to intensified OM deposition.*

In the revised version of the manuscript, we will further clarify how the sedimentation dynamics and OM deposition change in response to the gradual change in the physical configuration of the basin and to climatic oscillations.

**2. Responses to online discussion Referee #2**

**General Comments: The paper by Joniken et al. is a multiproxy study of the development of coastal hypoxia in the Archipelago Sea, Northern Baltic Sea during the last millenium. It takes into consideration the natural changes in the basin geomorphology and sediment transport, driven by climate variability, with the anthropogenic inputs, as causes of the emergence of near-bottom hypoxia in the area. One of the main findings is that coastal hypoxia started to occur in the early 1900's, and not in the mid-twentieth century, as other studies have suggested. The authors state that natural variability and trends were the main drivers of the emergence of hypoxia in the 1900's, but that eutrophication was the key factor from the 1950's onwards. The paper is well-written, including many data in tables and figures. Perhaps alternative interpretations on the studied record are missing, but, overall, the manuscript deserves to be accepted for publication. Minor revisions are suggested as follows.**

*We thank the referee for careful consideration of our manuscript. Below we address each of the specific comments.*

**Specific comments: 1) The'material and methods' section is too long and might be simplified. For example, subsection 4.4, could be summarized and the full text be moved to supplementary material.**

*We thank the referee for the suggestion. However, we find that our Materials and methods section follows the standard level of detail in the journal.*

We suggest that the decision about moving some parts of this section to the supplement is left for the editor.

**2) p.22 The authors propose that Mo accumulation rate is a proxy for bottom water hypoxia in the MoWP, suggesting that this takes place mostly during summer. Nevertheless, Mo geochemistry depends on the suplhide concentration in the pore waters, which in turn also depends on the sedimentation rate of labile organic matter. As the authors recognize, the phytoplankton productive season extends during spring/summer. Thus Mo accumulation rate does not actually depend on bottom water hypoxia, rather the latter be a consequence of organic matter respiration in the surface sediments. This should be taken into account in the discussion. On other hand, in order to preclude any bias due to the changes in the sedimentation rate for the authigenic accumulation of Mo, it would be better to normalize the Mo content to Aluminum, and estimate the Mo enrichment factor (Scholz et al., 2013).**

*The referee is correct that normalization against Al is a good practice. In fact, our Mo/Al data as well as the enrichment factor (EF) for Mo (calculated following Scholz et al. Chem. Geol. 355, 2013) both show exactly the same trends as the Mo content profile used in the manuscript (Fig. R2), implying that our raw Mo content reflects the trends in authigenic Mo sequestration in the sediment. Therefore, we conclude that the interpretation of the records remains unchanged whether we use Mo/Al ratio, Mo EF or raw Mo content. As we need the raw Mo content for the calculation of Mo MAR, we found it is more instructive to present the Mo content and MAR side by side in the manuscript instead of presenting the Mo/Al or Mo EF profiles. The referee is also correct that Mo sequestration*

*is modulated by the delivery of labile OM, which in turn drives the sulfide accumulation within pore waters upon*
*microbial degradation. This is actually mentioned on page 23 in our manuscript: "Another marked redox shift is*
*observed at 1950 AD, where a rapid increase in Mo MAR accompanied by a prominent decrease in Pr/Ph suggest*
*unprecedentedly reducing conditions at the sediment–water interface (Fig. 6). This shift likely denotes shoaling of*
*the redox zonation as a response to eutrophication in the area, which has been reported in previous studies of the*
*Baltic Sea (Slomp et al., 2013; Egger et al., 2015; Rooze et al., 2016: Jilbert et al., 2017) and reflects intensified*
*delivery of labile OM to the seafloor with respect to the supply of electron acceptors (Middelburg and Levin,*
*2009)." Yet, we agree that the first-order control of OM accumulation on Mo sequestration should be stated more*
*clearly in the manuscript.*

In the revised manuscript, we will clarify the role of labile OM delivery as a driver of sulfide accumulation and
subsequent Mo sequestration in sediment. We stand by the presentation of Mo content and MAR data instead of
Mo/Al and Mo EF for the reasons given above, but we will present the Mo/Al profile in the supplement to show
that the Mo content reflects authigenic Mo sequestration.

[Figure]

Fig. R2. Comparison of the raw Mo content profile with Mo/Al and Mo enrichment factor (EF).

**3) p.23 on causes of early 1900s deoxygenation: the authors include stratification and global warming as one of the factors precluding the bottom water ventilation and the deoxygenation trend. However, SST paleorecords (Figure 5), shows that the warming is significant (above that attained in the MCA) just around 1950, while laminations, increased d15N and increased organic carbon and Mo accumulation rates are recorded well before.**

*This relates to the same issue regarding the onset of hypoxia in the beginning of the 1900s that was pointed out by the referee #1. As already mentioned, we will put more emphasis on the combined effects of physical and anthropogenic forcing. As for the linkage between temperature and bottom water ventilation, we note that the warming trend in the dendroclimatic temperature reconstruction (Fig. 5) began already in the 1800s, whereas the instrumental temperature record (Fig. 6) unfortunately only extends to the turn of the 20th century. Although the referee is correct that the MCA temperatures were exceeded around 1950s, the warming trend that obviously started decades earlier likely contributed to the deoxygenation alongside anthropogenic forcing already in the beginning of the 20th century.*

Again, we will attribute the early onset of hypoxia (1900 AD) to the combined effects of physical and anthropogenic forcing, and stand by the inference that the warming trend in climate was one of the physical factors.

**4) The attribution to 'warmer climate' for the early deoxygenation is also mentioned in the conclusions. Authors might reconsider this interpretation or provide more support to this conclusion.**

*See our previous response.*

We retain the argument that the warming trend in climate was one the physical factors stimulating the onset of hypoxia around 1900 AD.

**5) p.20. On the role of the sewer network for the city which likely enhanced sewage loading to the Archipelago Sea, at the beginning of the twentieth century, the authors indicate that it had a secondary role, because no significant changes in the organic matter source were detected. However this argument is not convincing, since the increases of MAR, OC MAR and d15N occurred just after the sewer construction, and a higher influence of marine organic matter was recorded later, driving the exacerbation of hypoxia. It would be possible that anthropogenic organic enrichment partly sustained the early increase of MAR and also contributed the increase of primary productivity in the area.**

*Partly agreed. See our previous responses. Again we note that the long-term changes in the depositional setting as well as the constant C/N, BIT and $\delta^{13}C_{org}$ values over this regime shift around 1900 AD should not be ignored.*

As stated in the previous responses, we will modify the interpretation about the causes of the deoxygenation since 1900 AD to include anthropogenic forcing as one of the drivers alongside physical factors.

**5) p.25 ('Implications'). The authors indicate that the δ15N record demonstrates increased anthropogenic nutrient input already at the beginning of the 20th century. It would be worth to analyze if the record is also related to an increased denitrification in the bottom waters and surface sediments, which could result of an increasing anthropogenic/natural labile organic matter flux.**

*We thank the referee for pointing out the potential influence of possibly increased denitrification rate on the sediment $\delta^{15}N$ values. However, there are several lines of evidence suggesting anthropogenically-induced riverine input of heavy nitrate to be a more likely candidate for the increase in sediment $\delta^{15}N$ values since the turn of the $20^{th}$ century. Firstly, the pattern is similar in coastal sites around the Baltic Sea, irrespective of the bottom water oxygen levels (see e.g. Voss and Struck, Mar. Chem. 59, 1997; Struck et al., Mar. Geol. 164, 2000; Voss et al., J. Marine Syst. 25, 2000; Savage et al., Limnol. Oceanogr. 55, 2010; Ning et al., 2018), and therefore the progressive enrichment in sediment $\delta^{15}N$ values over the $20^{th}$ century is likely a manifestation of synchronous changes in land use and urbanization in different parts of the Baltic Sea catchment. Secondly, in the Swedish coastal areas of the Baltic Sea, the onset of clear laminations in the sediments is observed after 1950 AD (Persson and Jonsson, Mar. Pollut. Bull. 40, 2000; Savage et al., 2010), while the sediment $\delta^{15}N$ values show an increasing trend already since ~ 1850 AD. In fact, in the record from Himmerfjärden (Savage et al., 2010), there is a notable plateau in the $\delta^{15}N$ profile at the onset of clear laminations rather than steepening of the gradient. Similar pattern is also observed in the Gåsfjärden record (Ning et al., 2018). Thirdly, although aggravation of hypoxia in the Archipelago Sea has continued to the present (Fig. 7), it seems that the progressive increase in $\delta^{15}N$ signature has shifted to a decreasing trajectory (Fig. 6) suggesting decoupling between deoxygenation and sediment $\delta^{15}N$ values. Finally, the effects of aggravation of seasonal hypoxia in the coastal areas of the Baltic Sea are questionable. For example, in an incubation experiment the denitrification rate remained unaffected over a shift from oxic to anoxic conditions (Hietanen and Lukkari, Aquat. Microb. Ecol. 49, 2007). Further, the coastal N removal at a monitoring station in the Gulf of Finland was markedly lower during 2007-2009 in comparison to 2003-2004, which was possibly attributed to deoxygenation (Jäntti et al., Aquat. Microb. Ecol. 63, 2011). Based on these considerations, we infer that the progressive increase in $\delta^{15}N$ values of our sediment record was most likely linked to increased riverine input of isotopically heavy nitrate from anthropogenic sources.*

We stand by the inference that the riverine influx of isotopically heavy nitrate from urban and agricultural sources caused the progressive enrichment in the $\delta^{15}N$ values in our sediment record over the $20^{th}$ century.

**6) Figure 7. Though all the figures coincide in showing a negative trend of near-bottom water oxygenation; it would have been better to provide a climatology (or monthly/seasonal averages) of the water column dissolved oxygen concentration. The resolution of the time-series does not allow to observe the seasonal hypoxia.**

*We apologize it was not mentioned in the caption that these interpolations are based on water column oxygen concentrations in August. The purpose of this figure is to show the decadal-scale trends in oxygenation around the*

*Archipelago Sea, not seasonal dynamics. In addition, we note that the density of the monitoring data in the majority*
*of the monitoring stations does not allow high-resolution observation of the seasonal hypoxia.*

We retain the figure unchanged. We will mention in the caption that the interpolations are based on oxygen
concentrations in August.

**Technical comment: The legend of Figure 2 (geochronologies) is missing (the text of the legend 1 was**
**duplicated here).**

*We apologize for the inconvenience.*

This will be corrected in the revised version of the manuscript.

---

## Author Response (AR2)

**Contents:**

**1. Responses to online discussion Referee #1**

5 This document contains the responses to Open Discussion of "A multiproxy record of coastal hypoxia from the northern Baltic Sea indicates unprecedented deoxygenation over the 20th century" (second round of revision) and a description of the subsequent modifications in the revised manuscript, following the sequence described below.

**[Referee comments in bold]**

*[Responses in italics]*

10 [Changes in the manuscript in regular text]

**To my opinion the authors have improved the quality of the manuscript substantially. Previous critics have largely been addressed and I have only two small remarks:**

*We thank the referee for thorough consideration of our revised manuscript. Below we address each of the specific comments.*

15 **Page 18/19**

**N/C instead of C/N was written which should be corrected.**

*When referring specifically to the mixing model, we used N/C since the calculations were done based on N/C ratio instead of C/N for the reasons described in the Section 4.4.1.*

For simplicity, we have now modified the manuscript so that we consistently use C/N instead of N/C, except for 20 the Section 4.4.1.

**Page 27 line 6**

**It would be helpful for the reader to name the natural processes.**

*These natural changes related to the gradual evolution of the basin configuration are explained earlier in the Conclusions: "The progressive deoxygenation during the 1900s was originally triggered by gradual shoaling of*

[Figure]

*the basin due to glacio-isostatic uplift and basin infilling that, together with warming climate and anthropogenic nutrient input, promoted the vulnerability of the basin to hypoxia and intensified OM accumulation." Yet, we admit that clarification is needed to make to linkage more obvious.*

In the revised version, we have clarified the sentence pointed out by the reviewer.

**2. Marked up revised manuscript**

[revised manuscript text omitted]